# Potassium fertilization enhances both cereal yield and soil organic carbon: a meta-analysis

Guopeng Liang [1] ✉ & William H. Schlesinger [2] ✉

Agricultural ecosystems play a significant role in global food security and climate mitigation through crop production and soil organic carbon sequestration. It is well-established that potassium fertilization enhances crop yield in potassium-deficient regions; however, the factors driving crop yield responses to potassium remain insufficiently characterized at a large scale. Moreover, despite the significant roles of soil organic carbon in soil health and global carbon cycling, the effect of potassium on soil organic carbon in croplands has been less studied. Herein, we collect data from 1185 observations in agricultural ecosystems to conduct a meta-analysis study. We find that potassium fertilization increases cereal yield and soil organic carbon by 19.3% and 4.4%, respectively. Mean annual precipitation and experimental duration are the most important factors affecting potassium effects on cereal yield and soil organic carbon, respectively. Specifically, potassium effects on cereal yield increase with mean annual precipitation, and the potassium-induced increase in soil organic carbon is significant only after long-term (> 20 years) potassium fertilization. Our findings suggest that, in addition to nitrogen and phosphorus, potassium is also crucial for not only cereal yield but also soil carbon sequestration, which should be fully valued in future soil nutrient management, especially in potassium-deficient regions.

Chemical fertilization plays an important role in addressing the global food crisis. For example, nitrogen (N) and phosphorus (P) fertilizers can increase crop yield by 62%[1] and 14%[2], respectively, at the global scale. As one of the macronutrients and the most abundant cation in plant cells[3], potassium (K) is recommended to be applied together with both N and P for the maximum crop yield[1], particularly in soils where K availability is a limiting factor. When compared to N and P, however, the effect of K on crop yield and its controls are poorly quantified at the large scale[4,5]. In addition to crop production, agricultural ecosystems have the potential to slow global warming through sequestering carbon (C) in soils[6]. However, unknowns remain regarding how K can contribute to climate mitigation by improving C inputs (e.g., root production and crop residue) and thus soil organic C (SOC). The two knowledge gaps inhibit our comprehensive understanding of the roles of K in food security and climate mitigation, as illustrated below.

Potassium is crucial for crop growth since it is associated with plant photosynthesis and the synthesis of carbohydrates, proteins, and starches[3,7]. In addition, K can defend plants against various biotic and abiotic stresses (e.g., pests, diseases, salinity, drought, and waterlogging)[8]. Therefore, many studies found that K additions can significantly increase N use-efficiency and crop yield[9,10]. However, the extent of K limitation on crop yield is often regionally specific. It is estimated that approximately 18% of global agricultural lands face threats from K deficiency[11], with the most vulnerable areas concentrated in Southeast Asia, South America, sub-Saharan Africa, and East Asia[12]. Some recent studies from actual production systems confirm that K deficiency remains a severe constraint in farmers' fields across South Asia[13] and Indonesia[14]. Reflecting these regional needs, a substantial body of experimental research has been conducted in these K-limited areas over the past decades. This could provide a

[1]Department of Ecology & Evolutionary Biology, Yale University, New Haven, CT, USA. [2]Cary Institute of Ecosystem Studies, Millbrook, NY, USA.
✉e-mail: Guopeng.Liang@yale.edu; schlesingerw@caryinstitute.org

robust, albeit regionally-focused, database for synthesizing how cereal yield responds to K fertilization under conditions of nutritional stress.

Previous global meta-analysis studies found that crop yield was more limited by N[1] and P[2] in warm and wet than in cold and dry regions, and the effects of N and P on crop yield increased with N and P application rates[1,2], respectively. These results suggest that the response of crop yield to chemical fertilization should consider climate variables and fertilization management. However, uncertainties remain regarding the K limitation of crop yield. For example, to the best of our knowledge, no studies have been conducted at large scales to comprehensively identify other factors (e.g., climatic and soil variables and K application management) affecting crop yield. Furthermore, although temporal patterns of nutrient (e.g., N and P) effects on crop yield have been widely found[2,15], long-term effects of K on crop yield have been less studied. Addressing these knowledge gaps would provide useful information to help policymakers, landowners, and farmers apply the right fertilizer management for higher crop yield.

As an important indicator of soil health and fertility, SOC plays a fundamental role in not only improving crop yield but also fighting climate warming[6]. However, the effect of K on SOC has been overlooked for a long time. Potassium can be easily leached from soil organic matter[7], demonstrating that K does not directly interact with SOC cycling. Therefore, K effects on SOC should derive only from its effects on plant growth[16]. When the aboveground part of crops is harvested, the roots are usually left in the soil. Theoretically, K should indirectly increase SOC by improving root growth[17–19] and exudates[20–22] (the important C inputs to soils) in agricultural ecosystems. Given the great role of SOC in soil health, food security, and climate mitigation, it is essential to determine the indirect effect of K on SOC and identify the potential variables mediating K effects on SOC. This meta-analysis study aims to assess K effects on both crop yield and SOC and to determine what controls these effects. We hypothesized that 1) due to the high leaching rates of K under high mean annual precipitation (MAP), crop yield in wet regions should be more constrained by K limitation; and 2) given that crop roots can be fully decomposed in 0.5 to 2 years[23] and it takes relatively long to accumulate SOC even under straw return[24], the effect of K on SOC should be small in the short term but become significantly positive in the long term.

Here, we show that K fertilization enhances cereal yield and SOC by an average of 19.3% and 4.4%, respectively. Our meta-analysis reveals that the K-induced yield increment is positively correlated with mean annual temperature, precipitation, soil pH, and K application rate. Furthermore, the positive effect of K on SOC becomes significant only after long-term application (e.g., >20 years). These findings suggest that K fertilization represents a critical strategy, particularly in K-deficient regions, to synergistically improve crop production and soil C storage, thereby contributing to global food security and climate change mitigation.

## Results

### Potassium effects on cereal yield
Across all 897 observations in our meta-analysis, K additions averaging 83 kg K ha$^{-1}$ crop season$^{-1}$ increased cereal yield by 19.3% (95% CI: 14.7%–24.0%; Figs. 1a and S1a). Significant effects of K on cereal yield were found when K was applied with N or both N and P. In addition, cereal yield response to K was positive only when K type was KCl (+ 21.1%; 95% CI: 15.3%–27.2%) or $K_2SO_4$ (+ 16.2%; 95% CI: 6.1%–37.3%). We also found that K effects varied with the crop system. More specifically, K effects on cereal yield were greater in the rotation than the monoculture crop system. Positive effects of K on cereal yield were found for barley, maize, rice, and wheat (+ 16.5%; 95% CI: 2.4%–32.5%, + 21.3%; 95% CI: 14.3%–28.8%, + 12.1%; 95% CI: 3.9%–21.0%, and + 24.3%; 95% CI: 17.1%–31.9%, respectively), but not for millet and sorghum. Potassium effects on cereal yield were positive in short (0-10 years) and long-term (> 20 years), but were non-significant in the mid-term (10-20 years).

The positive effect of K on cereal yield was only found in China & East Asia (+ 12.9%; 95% CI: 6.6%–19.7%) and South & Southeast Asia (+ 30.8%; 95% CI: 20.6%–41.7%). Among all predictor variables, MAP was the most influential factor affecting K effects on cereal yield (Fig. 2a). This finding was also supported by the positive linear relationship between MAP and K effects on cereal yield (slope = 0.0002, $P < 0.001$; Fig. 3a), which is consistent with our first hypothesis. Overall, the response of cereal yield to K was mostly explained by soil (32.7%), climate (32.5%), fertilizer (27.8%), and crop variables (7.0%) (Fig. 2a). Moreover, we found that cereal yield response to K was negatively correlated with soil pH (slope = - 0.0797, $P < 0.001$; Fig. 3b), but was positively related to K application rate (slope = 0.0005, $P < 0.05$; Fig. 3c) and mean annual temperature (MAT, slope = 0.0080, $P < 0.001$; Fig. 3d). While the aforementioned correlations were statistically significant ($P < 0.05$), the values of $R^2$ were relatively low. This is common in the large-scale meta-analysis because the dataset encompasses a vast range of environmental gradients (e.g., diverse soil conditions, climatic regimes, and management practices), which contribute to the high residual variation. Non-significant linear relationships between the other predictor variables and the response ratio of cereal yield to K were found ($P > 0.05$; Fig. S2).

### Potassium effects on SOC
Overall, K increased SOC by 4.4% (95% CI: 0.9%–7.9%) in agricultural ecosystems, which were mainly located in South Asia and China (Fig. 1b). Potassium effects on SOC were positive when both N and P were also applied (+ 6.3%; 95% CI: 2.4%–32.5%), but became non-significant under other fertilizer combination scenarios. The positive effect of K was found for KCl (+ 6.1%; 95% CI: 1.3%–11.1%) but not for $K_2SO_4$ or mixed. In addition, K did not significantly affect SOC in the monoculture crop system, but a positive effect of K on SOC was found for the rotation system (+ 5.3%; 95% CI: 1.2%–9.7%). Potassium effects on SOC were significantly positive in C3 crops (+ 5.3%; 95% CI: 0.1%–9.3%) but became non-significant in C4 crops. SOC response to K was non-significant in the short and mid-term (e.g., <20 years) but became positive in the long term (e.g., > 20 years, + 4.4%; 95% CI: 0.5%–8.5%), which supports the second hypothesis. A positive effect of K on SOC was only found in South and Southeast Asia (+ 7.5%; 95% CI: 0.4%–15.1%). Experimental duration was the most important factor affecting K effects on SOC, which explained 21.4% of its variation (Fig. 2b). Non-significant linear relationships between the continuous predictor variables and K effects on SOC were found ($P > 0.05$; Fig. S3).

## Discussion

### Potassium fertilization increased cereal yield
The 19.3% increase in cereal yield observed in this study is comparable to the yield gains (ranging from 13.9% to 25.9%) reported for P fertilization[2], suggesting that optimized K nutrition is equally vital for enhancing crop yield. However, several caveats must be considered when interpreting these findings. First, as most experiments in this meta-analysis were conducted in K-limited environments, a direct extrapolation of this effect size to the entire global cropland may lead to an overestimation. Second, regional K application trends have shifted significantly over recent decades. In China, for example, the partial nutrient balance transitioned from a deficit of -14 kg $K_2O$ ha$^{-1}$ in 1980 to a surplus of +23 kg $K_2O$ ha$^{-1}$ in 2010[12]. Such a shift toward surplus suggests that the marginal yield returns from further high-rate K applications in these regions may diminish and are unlikely to consistently reach the 19.3% threshold observed in K-deficient trials. Furthermore, excessive K application in soils with adequate reserves can induce nutrient imbalances (e.g., magnesium deficiency)[4], potentially leading to yield declines. By contrast, in many least-developed countries, K deficiency remains unaddressed due to chronically low fertilizer inputs[12]. In these cases, targeted K fertilization offers a substantial, yet underutilized, potential to enhance cereal yield and alleviate local food crises.

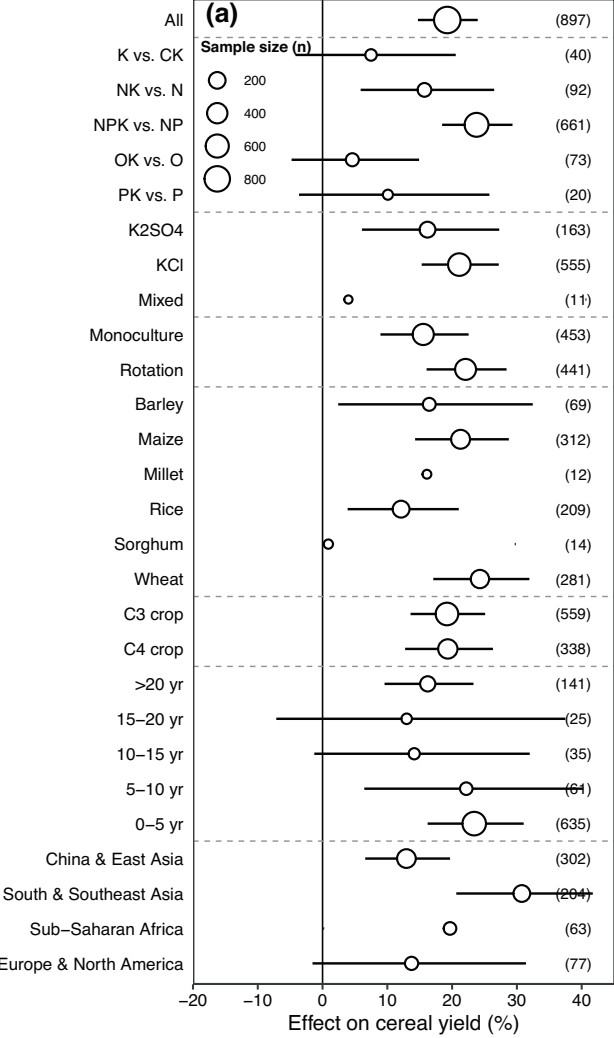
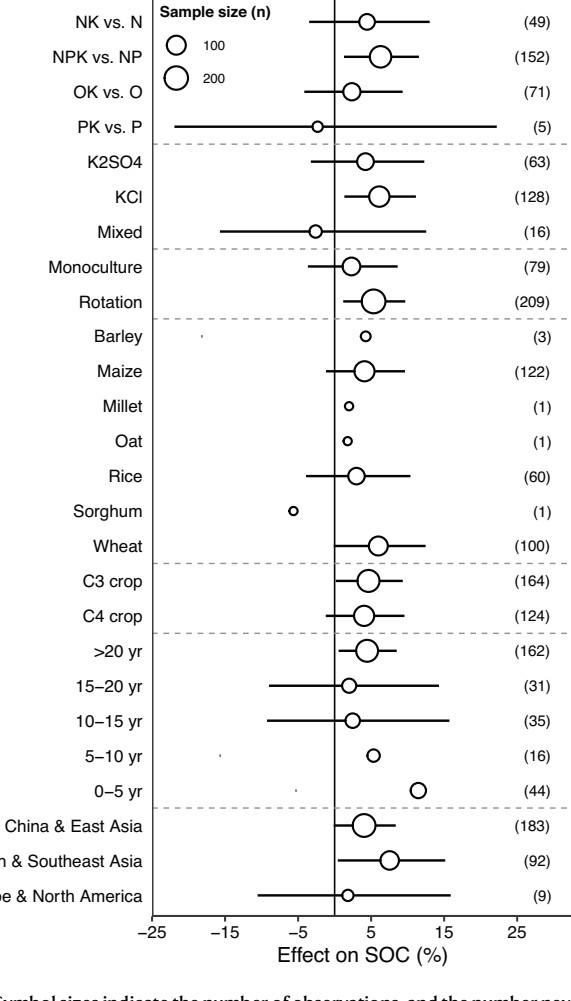

**Fig. 1 | Cereal yield and soil organic carbon under potassium fertilization.**
Effects of potassium (K) fertilization on cereal yield (**a**) and soil organic carbon (SOC, **b**). Results on the percentage changes (mean ± 95% CIs) in studies grouped by fertilizer combination, K fertilizer type, crop rotation system, crop type, and experimental duration. OK: organic fertilizer plus K fertilizer; Mixed: different K fertilizers combined. Potassium effects were significant if the 95% CIs did not cover zero. Symbol sizes indicate the number of observations, and the number next to the dots is the sample size of each variable. 95% CIs for Mixed, Millet, Sorghum, and Sub-Saharan Africa (cereal yield) and Millet, Oat, Sorghum, 5-10 yr, and 0-5 yr (SOC) were not displayed as they exceeded the axis range. Source data are provided as a Source Data file.

We found that MAP, soil pH, and K application rate were the three most important factors affecting cereal yield response to K, which explained more than 50% of its variation. Due to the great solubility of K in soils, K can be leached more rapidly from soils under high MAP[7,25], ultimately leading to K deficiency in wet regions, such as the humid tropics[5]. Because of the significantly negative relationship between soil pH and MAP found in this study, it is not surprising that K effects on cereal yield decreased with soil pH. Moreover, this result might be because the large amount of $H^+$, $Al^{3+}$, and $Mn^{2+}$ ions in acidic soils can significantly compete with K for uptake by plant roots[26], potentially reducing K absorption. All of these can result in a more serious K deficit under low soil pH; therefore, K addition would result in a greater increase in cereal yield under low soil pH. These results and the positive relationship between MAT and K effects on cereal yield suggest that, when compared to other climate regions, cereal yield in tropical zones (characterized by high MAT and MAP and low soil pH) is limited not only by N[1] and P[2] but also by K, which has been reported by some previous studies[12–14].

Although not among the three most influential factors affecting K effects on cereal yield, the role of fertilization management is crucial

because it can be adapted by policymakers and farmers more easily to improve cereal yield[1,27,28]. Potassium effects on cereal yield were significantly positive only when N or both N and P were co-applied, indicating that N should be applied with K for achieving higher crop productivity. Since organic fertilizers already supply a considerable amount of K, which partially fulfills crop requirements, the additional response to mineral K fertilization was not statistically significant when organic fertilizer was present in both the control and treatment groups.

**Potassium fertilization enhanced SOC**
The K-induced increase (+ 4.4%) in SOC suggests that applying K fertilizer in regions with K deficiency can be a promising nutrient management strategy for soil C storage. It should be noted that experiments determining K effects on SOC are mainly conducted in China and India. Moreover, almost none of the studies included in this meta-analysis measured soil nutrient leaching rates and soil C emissions under K addition. To more accurately estimate the response of soil health and C cycling to K fertilization at a global scale, future research should be expanded across diverse geographical regions, and

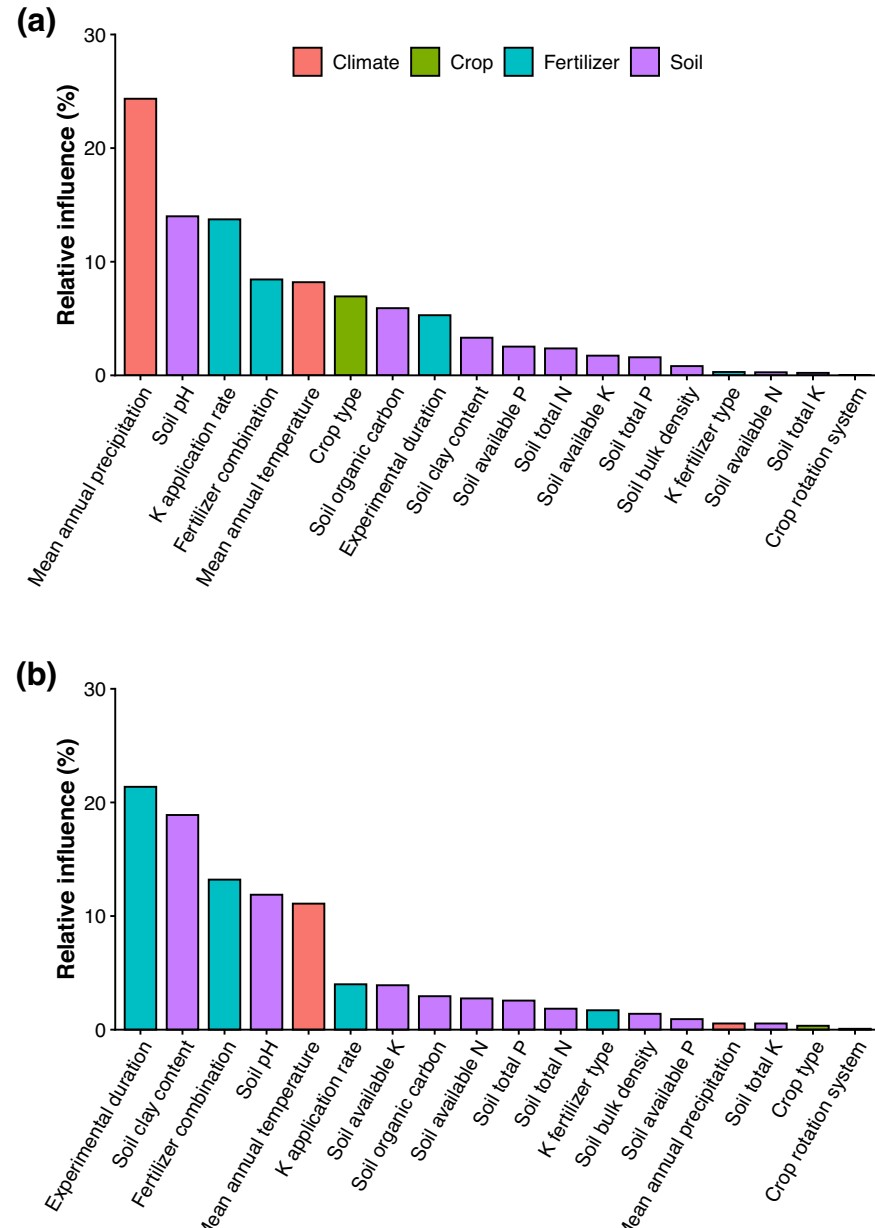

**Fig. 2 | Factors affecting the responses of cereal yield and soil organic carbon to potassium fertilization.** The relative importance of predictor variables of potassium fertilization effects on cereal yield (**a**) and soil organic carbon (**b**). N: nitrogen; P: phosphorus; K: potassium. Source data are provided as a Source Data file.

a broader suite of soil health indicators needs to be systematically evaluated.

Key variables underlying soil C cycling (e.g., root biomass and exudates) were rarely reported in the studies included in this meta-analysis. This lack of data precludes a mechanistic understanding of how K fertilization drives positive SOC responses. However, our findings that experimental duration and fertilizer combination were among the three important factors affecting K effects on SOC have some important implications for farmers and policymakers, especially in K-deficient areas. First, our results indicate that a significant increase in SOC usually emerges after approximately 20 years of K application. While such a long-term commitment may pose challenges for short-term agricultural planning, it highlights the importance of sustained nutrient management in building soil capital. This timeframe suggests that K fertilization should not be viewed merely as a transient input, but as a component of long-term sustainable soil management that yields cumulative benefits for both crop productivity and soil C

storage over time. Second, K should be applied together with both N and P to maximize the K-induced increase in SOC. Moreover, since K-driven SOC increases were specific to KCl treatments, fertilizer type is a key determinant for enhancing soil C storage through K addition.

## Methods
### Data collection
We used the Web of Science to conduct a literature search in November 2025. Keywords "potassium fertilizer AND crop yield" and "potassium fertilizer AND soil carbon" were used to search for papers reporting K effects on cereal yield and SOC, respectively. The following criteria were used to select papers: (1) studies were conducted at field sites in agricultural ecosystems; (2) to remove the potential effects of N and P, the difference between K treatment and control should derive only from K addition (e.g., K *vs.* control; NK *vs.* N; PK *vs.* P; NPK *vs.* NP; and organic fertilizer [O] +K [OK] *vs.* O); (3) only yield of cereal, which plays a great role in global food security, was included in the present

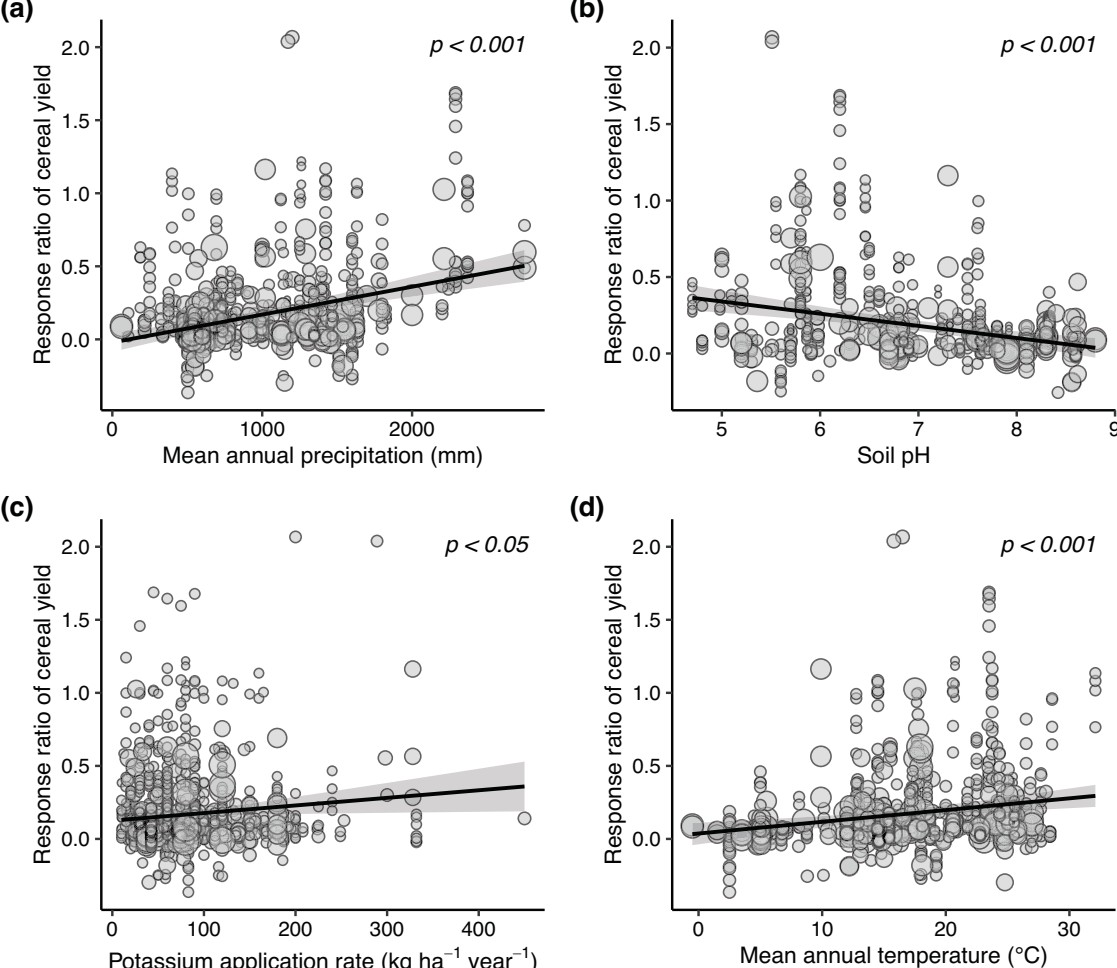

**Fig. 3 | Relationships between the response ratio (RR) of cereal yield to potassium fertilization and the predictor variables.** The solid black lines and shaded areas represent the predicted regression trends and their 95% confidence intervals, respectively. Circle sizes denote the statistical weight of individual studies. Statistical significance was evaluated using a random-effects meta-regression model, and the significance of regression coefficients was determined using a two-sided Z-test ($P < 0.05$) without adjustments for multiple comparisons. The $P$ values are 0.0000, 0.0000, 0.0278, and 0.0002 for (**a**)–(**d**), respectively. Source data are provided as a Source Data file.

study; (4) if SOC and other soil parameters were reported at multiple soil depths, the data in surface soils (within 20 cm depth) were used; (5) for studies reporting results at multiple time points, only the data at the last time point was included; and (6) cereal yield or SOC from different crop types, tillage systems, or K application rates in the same study were regarded as independent observations.

897 paired observations from 163 field sites and 288 paired observations from 75 field sites were collected to assess K effects on cereal yield and SOC, respectively (Fig. S4, Supplementary Data 1). The geographical distribution of our study sites is inherently linked to regional soil nutrient status. For example, fertilization experiments are seldom conducted in areas with adequate K reserves. Consequently, our database is characterized by a deliberate focus on regions with documented K limitations (e.g., East Asia and South Asia), especially concerning SOC data (Fig. S5). As highlighted by a previous study[11], these regions represent the primary global hotspots for K deficiency, making them the most relevant areas for assessing the potential of K fertilization. We also extracted site location, mean annual temperature (MAT) and precipitation (MAP), fertilizer combination (K vs. control; NK vs. N; PK vs. P; NPK vs. NP; and OK vs. O), K application rate, K fertilizer type (K₂SO₄, KCl, and Mixed [different K types combined]), experimental duration, crop rotation system (monoculture and rotation), crop type (barley, maize, millet, rice,

sorghum, wheat, and oat), and initial soil characteristics (SOC, pH, soil bulk density, clay content, and total and available N, P, and K) (Table S1). Total N, P, and K were defined as the whole-soil elemental concentrations determined via complete digestion. Available N, P, and K were defined as the plant-available fractions extracted using standard chemical reagents (e.g., alkali-solution or salt extractions for available N, Olsen or Bray methods for available P, and ammonium acetate extractions for available K). If the studies did not report MAT and MAP, we extracted them from the database at http://www.worldclim.org/, using latitude and longitude. Engauge Digitizer software (Free Software Foundation, Boston, MA, USA; Version 12.1) was used to extract the data in figures.

**Data analysis**

The following equation was used to calculate the natural log-transformed response ratio (lnRR) to quantify K effects on cereal yield or SOC:

$$\text{lnRR} = \ln(\bar{X}_t / \bar{X}_c) \tag{1}$$

where $\bar{X}_t$ and $\bar{X}_c$ are the mean values of cereal yield or SOC under K fertilization and control (no K fertilization) treatments, respectively. For studies reporting multiple K application rates, each rate was

extracted and treated as a separate observation to reflect the response across various K levels.

Most studies did not provide the standard deviation or standard error (643 of 897 studies and 182 of 288 studies for cereal yield and SOC, respectively). By using the following equation[29], we estimated the weighting factor ($W$) that was used to calculate the weighted response ratio ($lnRR_{++}$):

$$W = (n_c \times n_t)/(n_c + n_t) + (yr \times yr)/(yr + yr) \tag{2}$$

where $W$ is the weighting factor; $n_c$ and $n_t$ are sample sizes under control and K fertilization treatments, respectively; and yr is the experimental duration of the study in years. If the sample size was not provided in papers (66 of 897 studies and 42 of 288 studies for cereal yield and SOC, respectively), it was assigned as the median of the sample size of the other studies, which was 3 for both cereal yield and SOC.

We calculated the percentage change (%) of K effects and the corresponding 95% CIs as follows:

$$\text{Percentage change} = (e^{lnRR_{++}} - 1) \times 100 \tag{3}$$

where $lnRR_{++}$ is the weighted lnRR. Potassium effects were statistically significant when the 95% confidence intervals did not cover zero.

The "funnel" function in the package "metafor" in R was used to create a funnel plot, which is a graphical tool for visualizing publication bias (Figure. S6). Moreover, Egger's regression test (the "regtest" function) was performed to test funnel plot asymmetry. The $P$ values of Egger's regression test are 0.08 and 0.69 for cereal yield and SOC, respectively, indicating no publication bias. We then used the package "metafor" to perform the meta-analyses[30]. To quantify the relative influences of predictor variables on K effects on cereal yield or SOC, the package "gbm" was used to perform the boosted regression tree (BRT) analysis. Parameter values used for the BRT analysis, such as shrinkage, interaction.depth, and ntree were set as 0.005, 2, and 12000, respectively. The BRT analysis explained 82% and 72% of the variation in the response ratios (lnRR) of cereal yield and SOC. The function "rma" of the package "metafor" in R was used to determine the relationships between lnRR and all predictor variables. All statistical analyses and graphs were conducted in RStudio (Version 2024.04.0 + 735).

### Reporting summary
Further information on research design is available in the Nature Portfolio Reporting Summary linked to this article.

## Data availability
The datasets that support the findings of this study are openly available in Zenodo at https://doi.org/10.5281/zenodo.18839011[31]. Source data are provided with this paper.

## Code availability
The code that supports the findings of this study is available in Zenodo at https://doi.org/10.5281/zenodo.18839011[31].

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

## Acknowledgements

We thank all the researchers whose data were used in this meta-analysis study. Special thanks are extended to Pengyan Sun, Research Intern in the Department of Ecology and Evolutionary Biology at Yale University, for her diligent efforts in data collection.

## Author contributions

G.L. conceived and designed the study, analyzed the data, and wrote the first draft. W.H.S. was involved in writing and editing subsequent drafts.

## Competing interests

The authors declare no competing interests.
