## [Peer Review file · Nature Communications]

Potassium fertilization enhances both cereal yield and soil organic carbon: a meta-analysis

Corresponding Author: Dr Guopeng Liang

Version 0:

Reviewer comments:

Reviewer #1

(Remarks to the Author)

The paper describes a meta-analysis of the effects of potassium (K) fertiliser on yield and soil organic carbon (SOC) and concludes that K fertilisation improved yields of a range of cereals by 18% and SOC by 4%. The analysis is timely as there is some evidence of increased frequency of K deficiency in many crops in a number of countries and it highlights the potential benefits of improved K nutrition. However, I feel there is a tendency for extrapolation beyond the limits of the data. To say that a good supply of K is important for productive agricultural systems is without question, if only because K is the second most required nutrient after N for plant growth. However, this does not automatically lead to a conclusion that global cereal production will benefit from better K nutrition. The problem can be regionally-specific. For example, based on a 2001 analysis by Wood et al (2001), it has been estimated that 18% of the world's agricultural lands have soil with low K reserves and are threatened by K deficiency, with the largest areas in SE Asia, South America, sub-Saharan Africa and East Asia. However, there are other regions of the world where low soil K reserves are minor or not a problem.

The locations of the studies from which the data were drawn are concentrated in East Asia and South Asia and, to a lesser extent, sub-Saharan Africa, with relatively fewer studies in other regions. This is especially the case with the SOC data. This is not surprising given these regions have been identified in the past as areas with a high frequency of soils with low K, and it would be useful to acknowledge the earlier analysis by Wood et al (2001) that indicates this.

The analysis found an average 18% increase in yield and a 4% increase in SOC on a global scale. While the interpretation of the analysis is correct, there needs to be some qualification to the interpretation, otherwise there is a tendency to exaggerate the effect of K. The inference from the manuscript's title and parts of the discussion is that K deficiency is widespread throughout the world and most cereals crops suffer from some degree of K deficiency (line 131:most cereals are K limited at the global scale); this is not the case. The analysis is heavily focussed on responses in regions where there are soils low in K and where K fertiliser experiments have been conducted. Fertiliser experiments will generally not be conducted where K deficiency has not been diagnosed or is suspected to occur, and the fact that the data base is heavily biased to the regions with soils with low reserves of K is not coincidental. Applying K fertiliser to soil that has adequate K will not result in a yield increase, and in some cases may result in a yield decline from nutrient imbalances, such as Mg deficiency (Romheldt and Kirkby 2010). What the analysis shows is that in regions where K deficiency is likely to occur, the average yield response to K is 18%. This is a more nuanced interpretation of the data than arguing that most of the world's cereal crops are deficient in K with an average yield response of 18%. A similar argument regarding the 'global' effect of K on SOC applies.

Consequently, the estimate of the global increase in production (lines 178-179) overestimates the effect of K as it assumes the 18% increase will occur in all crops irrespective of whether they are K deficient or not. The regions of the world with the highest proportion of soils with low K (SE Asia, South America, Sub-Saharan Africa and East Asia) produced 1.4 billion tonnes of cereal grain (wheat, rice, maize, sorghum, barley, oats and millet) in 2023 (based on FAO data), so a more realistic estimate of the potential increase in production with K fertiliser is 0.25 billion tonnes, about 40% of the estimate given in the paper.

The other assumption implicit in the analysis is that the levels of K deficiency found in fertiliser experiments are representative of the severity found in farmers' crops. Over the last 15 years there has been an increase in K applications in many of the areas at risk of K deficiency, which raises the question of whether the extent of K deficiency is as great as inferred

in the paper. For example, FAO fertiliser use data shows that in East Asia the average rate of K application for all agricultural use (i.e. crops, horticulture, pastures etc) increased from 15 kg K/ha to 29 kg/ha and the K:N ratio of fertiliser used also increased from 0.08 to 0.15. However, in the Least Developed Countries, the changes have been much less and rates much lower, with average rates increasing from 0.6 kg/ha to 1.7 kg/ha and the K:N ratio increasing from 0.07 to 0.11, suggesting K deficiency may not be addressed in these countries. In China the partial nutrient balance changed from an average deficit of -14 kg K₂O /ha in 1980 to a surplus of +23 kg K₂O /ha in 2010 (Majumdar et al 2021). These examples illustrate that discussions of global impacts of K mask the regional nature of the problem. The Introduction would be improved if the wider evidence for K being a limiting factor in farmers' production systems was considered so the link between experimental results and farmers' fields is clearly established. The regional nature of the problem could be mentioned. For example, recent papers by Srinivasarao et al (2023) for South Asia, Rizzo et al (2024) in Indonesia document the extent of K deficiency while Majumdar et al. (2021) provide analyses of K balances for a range of countries. The conclusion from the analysis that K responses were greater in tropical regions is consistent with many of these previous analyses.

It would be useful to summarise the distribution of the data for the increases in yield and SOC by including a figure showing the frequency distribution of lnRR for yield and SOC as supplementary information. While the analysis assumes a normal distribution of the lnRR data, looking at the data set provided as supporting information suggests some skewness in the (unweighted) yield response with a median lnRR for yield of 0.0472; this is equivalent to a 4.8% yield increase, substantially lower than the mean 18.4% used in the analysis. The figure would illustrate whether there is any significant skewness to the data.

Line 12. Potassium will increase crop yields when crops are K deficient. The sentence suggests an increase will occur irrespective of the degree of K deficiency.

Line 84. There was a positive response to K in millet. Figure 1 shows a yield increase in of about 25% from three samples

Figure 3. While the correlations are significant because of the large data set, the amount of the variation explained is small.

Line 94. 'K increased SOC in agricultural ecosystems by 4.1% across the world' is perhaps exaggerating the extent of the results. Supplementary figure 6 shows there are no SOC data from North or South America, from most of Africa, Europe and SE Asia and from Oceania. The results are based on experimental largely from South Asia and China.

Lines 94ff. There is some contradiction in the description of the results which are not consistent with the statistical descriptions. For example, the description states that there was no significant variation among fertiliser combination on SOC but then describes that there was only a positive effect when both N and P were added (line 96-97). There was no significant variation in the effect on SOC among crop system, but later states that monoculture did not significantly affect SOC whereas there was a positive effect of crops grown in rotation (line 98). There was no significant variation in response in SOC among the different types of K fertiliser, but later it was stated that KCl was the only fertiliser that resulted in a positive response (line 98). If there is no significant difference amount different factors on SOC, you can't then argue that one factor has a larger effect than the other.

Line 102 and elsewhere. Statistical relationships are described as non-significant rather than insignificant.

Line 131. To say that 'most cereals are K limited at a global scale' is unsupported by the data set and exaggerates the effects for reasons stated previously.

Line 161-162. It is argued that the effects of MAT, pH and clay % on SOC were similar to their effects on yield, but this is not the case. While MAT, pH and clay content were the three most influential factors associate with the change in SOC (Fig 2b), only pH was important for yield. MAT was seventh and had a relative influence of 5% and clay content was 11th with a relative influence of about 1%

Line 181. While K fertilisation can be useful when K is deficient, it needs to be part of a balanced nutrient program and not as a stand-alone solution. This may have been the intention of the statement, but it is not well made.

Line 183ff. The potential to increase SOC from the use of K fertiliser needs to be viewed in the context of the rate of change in SOC. The analysis found that there was only a significant improvement in SOC in long term experiments which lasted more than 20 years (line 100, Fig 1). Therefore, K fertilisation is a long-term strategy and any short-term benefits will be small or non-existent.

Line 208. The analysis has quantified the effect of K on a regional scale not a global scale.

Line 210. It is a large jump to conclude that the changes in SOC that may occur with increased production from improve K fertilisation will slow down future global warming. There needs to be some quantification of the likely effect to justify the statement rather than an assertion that it has the 'potential' to do so.

Line 244-245. Explain what you define as total N, P and K and available N, P and K. There is a variety of extraction methods used to estimate soil nutrient concentrations, especially for available nutrients, and they can give different results.

What was the rationale for dividing the SOC data into experiments < 20 years and > 20 years? If the effect of duration was of interest would it be better to use more time periods?

Supplementary Fig 1. Are many of the non-significant relationships due to the sparse data set and the need to use the medians of the other data sets? With such poor representation, how relevant are these analyses?

While it was acknowledged in the text (line 104), many responses to fertiliser and to soil nutrient availability are curvilinear responses, so examining simple linear relationships can mask some of the responses. This seems to be the case for soil available K and to some degree K application rate.

References

Majumdar, K. et al. (2021). Assessing Potassium Mass Balances in Different Countries and Scales. In: Murrell, T.S., Mikkelsen, R.L., Sulewski, G., Norton, R., Thompson, M.L. (eds) *Improving Potassium Recommendations for Agricultural Crops*. Springer, Cham. https://doi.org/10.1007/978-3-030-59197-7_11

Rizzo G, Agus F, Susanti Z, Buresh R, Cassman KG, Dobermann A, Agustiani N, Aristya VE, Batubara SF, Istiqomah N, Oberthür T, Pasuquin J, Samijan, Witt C, Grassini P. Potassium limits productivity in intensive cereal cropping systems in Southeast Asia. *Nat Food*. 2024 Nov;5(11):929-938. doi: 10.1038/s43016-024-01065-z. Epub 2024 Oct 22. PMID: 39438701.

Römheld, V. & Kirkby, E. A. Research on potassium in agriculture: needs and prospects (2010). *Plant Soil* 335, 155–180.

Srinivasarao Ch, Kundu S., Rao KV, Shukla AK, Subba Rao A, Imas P., Bolan NS, Rattan Lal, Prasad JVNS., Abhilash PC, Ranjith Kumar G., Meena RS, Pratibha G., Narayanaswami G., Bansal SK, Nataraj KC, Jagadesh M., Mrunalini K., Jayaraman S., Jat ML, Malleswari SN, Whitbread A, Venkateswarlu B. (2023) Chapter Two - Soil potassium fertility and management strategies in South Asian agriculture. *Advances in Agronomy* 177:51-124.

Wood, S., Sebastian, K., & Scherr, S. (2001). *Pilot Analysis of Global Ecosystems: Agroecosystems*. International Food Policy Research Institute and World Resources Institute, Washington, DC

Reviewer #2

(Remarks to the Author)

This paper addresses an important and intriguing topic of K fertilization effects on global crop yield and soil organic C sequestration. The paper has clear objectives, methods, and results. The paper has potential to make important contributions to scientific literature and global policies. However, there are several issues that need to be addressed prior to publication. Foremost among these issues are the following:

1. The authors use questionable methods to estimate missing data, which has the potential to introduce substantial error into the dataset. Missing data should be left as missing.
2. The dataset has considerable geographic bias, which calls to question its utility for making a “global” assessment.
3. The authors’ description of study implication on global food supply may lead to misinterpretation of the data.
4. Authors are inconsistent with their interpretation of results.

A confounding factor with the data for this study is that researchers are more likely to conduct K-response trials on soils that have a higher probability of demonstrating K-response. Therefore, it is difficult to conclude that these studies represent a random selection of soils and potential K response throughout the world. Because this is not a random selection of K response, it would contain bias toward soils with probable K response and would likely be an over-estimate of the global effects of K fertilization. The authors should address this potential bias with the data.

Additional details relating to the above issues are listed in the detailed comments below, preceded by the line number(s) and followed with “issue number” from the list above if applicable.

75-77: The size of the 95% CIs appear to be directly proportional to the number of studies. Is the conclusion in these lines simply because there are far fewer studies without N&P, therefore, those confidence intervals are much larger and therefore do not differ from confidence intervals for N+P?

111-112: The statement “...indicating that K fertilization could further stimulate cereal yield...” appears to insinuate that K fertilizer is not being applied and if farmers applied K then it would increase world cereal yield by 18.4%. This is a misunderstanding (or could lead to a misunderstanding) of the data. These data indicate the yield difference for soils receiving K compared to if they did not receive K. However, many farmers already apply K. Therefore, we cannot “further” increase global yield by 18.4%, rather global yield with K is 18.4% greater than yield without K. The “further” increase would require data on the amount of K responsive soils that are not currently receiving K, which is not part of the study. Please revise. [Issue 3]

130-131: This data does not support the idea that most cereals are K-limited at the global scale. Remove statement. [Issue 3]

141 – 142: This is an incorrect conclusion because authors make the false assumption that if N or P fertilizer were not applied then the grain yield was N or P limited. It is highly likely that studies on soils with adequate N and/or P would not apply additional N or P when studying K response, therefore, one cannot assume that the lack of N or P fertilizer addition equates to N or P limitation. Please revise.

145-152: The authors' explanation for the lack of significant yield difference for the OK vs O comparison misses the strong possibility that K is applied with the organic fertilizer, therefore, no additional K would be needed and one would not expect a response to K fertilizer. Authors should determine the K addition with O for these studies the same as they did for N and P addition for O. Please consider revision.

151-152: the statement "...but led to the greatest increase in cereal yield when both N and P were applied as chemical fertilizer (Fig. 1a)." is contradictory to prior statements where the authors state that K response was the same regardless of N or P fertilizer addition (lines 75-77 and 141-142). Please revise. [Issue 4]

174 to 175: the statement "...in addition to contributing to terrestrial C sequestration in natural ecosystems, applying K fertilizers..." would suggest that K fertilizer is being or should be applied to natural ecosystems. Remove or revise this sentence.

178-180: Similar to comments for lines 111-112, this statement implies that K fertilizer application is not occurring. Because farmers are already applying K, the world cereal yield is realizing the benefits of K application. Therefore, more K application would not increase world production by 18.4%, but it would only increase production on the K deficient soils that are not already receiving K. Authors do not have any information about the fraction of K-responsive world soils that are not receiving K. Please revise. [Issue 3]

186: Similar to prior comment – K is already being applied, so we do not know the "additional" C sequestration that could be attained by applying more K. Please revise.

192 -194: These relationships are flawed because the authors did not have sufficient data on available K in the soils (see comments below). Therefore, the implications are misleading. Please repeat the analysis with revised data (see below). [Issue 1]

200 -201: Authors previously stated that N&P addition did not increase the K response (lines 75-77 and 141-142). Authors need to be more consistent when interpreting results. Please revise. [Issue 4]

201-203: Where is the data that would suggest "greater" response than what the authors found? Please revise.

203-205: Discussion on decreasing N application rates is beyond the scope of the study and authors do not present any data to support this discussion. Please remove these sentences.

233-244: The authors should provide a list of citations for all the papers that contributed data to the meta-analysis so that future studies can build on this study. Furthermore, the data presented in Figure S6 show extreme geographic bias for this dataset, especially for K effects on soil C, where there are only 3 studies outside of the Asian continent. Based on the extreme geographic bias, the authors cannot claim that this is a global representation of soil C response to K fertilization and it is highly questionable if they can make the claim that it is a global representation of the crop yield response to K fertilization. I am frankly quite surprised that the authors could not find more data on crop yield response to K in North and South America. It is hard to tell with the scale of the figure, but it appears that there are only 3 papers from the corn belt region of USA. If the authors cannot find more globally representative data then they will need to clearly include an acknowledgement of the geographic bias of the data within the results section of the paper. [Issue 2]

242-244: I doubt that this database could provide accurate information on dynamic soil properties that can easily be affected by field-level management, such as pH, total N, and SOC. Even bulk density of the field site may be drastically different than that listed in the database. If the authors feel otherwise, they could easily test this with the data from studies that do report these properties, which could provide prediction confidence interval that could be reported with the study. Authors should clearly indicate the number of studies using estimated data from this database in Table S1. [Issue 1]

244-246: The use of median values for total P, total K, available N, P, and K really have no relevance and will introduce massive error in the dataset. If the data are not available, then they should be marked as missing data. Note that the effects of replacing missing data with median values are obvious in Figure S2 and likely confound the results of many of these relationships. Please repeat the analysis without the estimated values. [Issue 1]

Figure 2: align the x-axis category names with the bars

Supp. Table 1 – This table must be updated to accurately reflect the data used in the study. The table shows N=744 for all cereal yield studies and indicates that N=744 for each of the response variables. Suggesting that every study had complete data. However, the authors clearly indicated that that data was not available for each soil property and that missing data was replaced by the median (lines 244-246). Please revise the table and clearly indicate the number of studies that reported values for the parameters listed in table S1. [Issue 1]

Reviewer #3

(Remarks to the Author)

This study utilized two separate databases to investigate the impact of potassium fertilizer application on soil organic carbon and crop yield. Compared to conventional research focusing on nitrogen and phosphorus fertilizers, the topic of this study is interesting. However, the study suffers from the following issues, which undermine the credibility of its results:

1. Although both databases focus on global field experiments, the distribution of sampling sites does not cover all typical global climate zones, particularly for soil organic carbon. This significantly affects the discussion of the results and the representativeness of the data.
2. Line 154-160: The authors assume a positive correlation between crop yield and biomass, inferring an increase in root biomass from crop yield improvement. This approach is insufficiently rigorous and not accurate, as no universal relationship necessarily exists between crop yield and biomass.
3. It is inappropriate to use crop yield data to explain soil organic carbon data, given that the data sources and involved climate zones for these two aspects differ substantially.
4. The study concludes that potassium fertilizer application can increase crop yield and promote carbon sequestration, but it fails to comprehensively consider the associated impacts. For instance, issues like leaching mentioned by the authors, and potential carbon emissions during the potassium fertilizer production process are not fully accounted for. Therefore, determining the most suitable fertilizer involves a need for balance and comprehensive consideration. The authors should also express certain viewpoints more carefully; for example, in line 212, the paper suggests to "applied continuously for a long time in order for a significant increase in SOC sequestration." Is this solution possible or acceptable for most places?
5. Without access to the authors' raw data, it is impossible to assess the accuracy of the data. For example, when the authors mention collecting "soil carbon" (Line 222), it is unclear whether this refers to total carbon or specifically to organic carbon. Furthermore, in Figure S1 of the supplementary information, the content of many indicators is concentrated around specific values (e.g., numerous data points for total soil phosphorus content near 0.6 g kg⁻¹). This raises the question of whether these data originate from a single publication, which could significantly impact the reliability of the paper's dataset.
6. Line 220: The exact date should be provided here. It has been nearly two years since 2023. New researches have been published, so the authors need to update the database to reflect the latest results and trends.
7. This study uses estimated weighting factor and the median of the sample size. As a quantitative analysis based on large databases, the accuracy and reliability mostly depend on the source of the data. Therefore, I doubt about the way the data is handled.

Version 1:

Reviewer comments:

Reviewer #1

(Remarks to the Author)

I thank the authors for their detailed response to the comments made on the previous version of the manuscript. They have addressed the major concerns I had with the original paper and the changes that have been made have resulted in the manuscript providing a more realistic interpretation of the data set. By acknowledging the data mainly relate to regions where the risk of K deficiency is highest, the analysis provides an assessment of the likely gains that can be achieved from improved K nutrition in these regions. The regions that are the focus of the paper already have been identified as areas where the risk K deficiency is relatively high and the issue has been discussed previously; in this respect the significance of the paper has diminished compared to a 'global perspective', but the analysis provides some new information on the magnitude of the likely yield benefits from correcting K deficiency and some of the factors that influence the responses.

Line 214. Although it is implied, it may be useful to define X_t as the maximum yield achieved with K fertilisation. If any of the data sets used in the analysis are based on K response curves, there will be a number of yields that will satisfy the current definition of X_t as 'under K fertilisation'. While it seems trivial, it will remove any ambiguity in the description.

While the authors have undertaken quite a comprehensive analysis, perhaps they could consider two further aspects that may provide further insights:

(i) In the analysis, 'cereals' is a combination of C3 and C4 species. Is there any evidence from the analysis that C3 and C4 species differ? The means for wheat and barley in Fig 1a are slightly larger than the C4 cereals but the CIs overlap. It may be worth commenting whether there is/is not a difference

(ii) To extend the concept of the regional nature of K deficiency, would it add value to the analysis if you included different geographic areas to see if there were marked differences in responses between for example, Sub Saharan Africa, China and South/SE Asia? This could then be related to the earlier analysis of Woods et al, and Majumdar et al

In general, sentences don't start with an abbreviation; for example, lines 35 and 102 should start with 'Potassium....', not 'K...', but I will defer to the editorial guidelines.

Line 27-28. "... potassium (K) is recommended to be applied together with both N and P for the maximum crop yield. The statement needs to be qualified. This will only be true if there is a chance of a response to K; there is no point applying K if there is adequate soil K.

Line 146. 'Although this yield trend did not reach statistical significance across groups.....'. It is unclear what this means; is it that the mean was not significantly different to the other fertiliser treatments (based on overlap of the 95% CI?). Please

clarify.

Lin 150-52. 'Given that organic fertilizer also provides K to crops, which can alleviate K limitation to some degree, it is not surprising that K effects on cereal yield were non-significant when organic fertilizer was applied. It is unclear to me what this means; it seems self-contradictory

Some of the references in the reference list are undated. Please include all dates of publication.

(Remarks on code availability)

Reviewer #2

(Remarks to the Author)

The authors have done a fabulous job of responding to comments. They have addressed all concerns and the paper is ready for publication with the exception of one small edit listed below.

line 78: Change "1185" to "897". Perhaps 1185 is the sum of 897 (yield dataset) and 288 (SOM dataset). However, because this statement is discussing yield, the authors only have 897 observations. This would also agree with figure 1a that is cited at the end of the sentence, which lists 897 observations.

(Remarks on code availability)

The readme file did not provide adequate metadata for the data files. Units for the data in data files was still unclear. Readme file did not contain any instructions for running the code.

REVIEWER COMMENTS

Reviewer #1:

The paper describes a meta-analysis of the effects of potassium (K) fertiliser on yield and soil organic carbon (SOC) and concludes that K fertilisation improved yields of a range of cereals by 18% and SOC by 4%. The analysis is timely as there is some evidence of increased frequency of K deficiency in many crops in a number of countries and it highlights the potential benefits of improved K nutrition. However, I feel there is a tendency for extrapolation beyond the limits of the data. To say that a good supply of K is important for productive agricultural systems is without question, if only because K is the second most required nutrient after N for plant growth. However, this does not automatically lead to a conclusion that global cereal production will benefit from better K nutrition. The problem can be regionally-specific. For example, based on a 2001 analysis by Wood et al (2001), it has been estimated that 18% of the world's agricultural lands have soil with low K reserves and are threatened by K deficiency, with the largest areas in SE Asia, South America, sub-Saharan Africa and East Asia. However, there are other regions of the world where low soil K reserves are minor or not a problem.

Response: Thank you so much for the comprehensive and constructive feedback. We completely agree that the yield and SOC response to K fertilization is highly context-dependent and regionally specific. We acknowledge that the initial version of our manuscript may have inadvertently over-extrapolated the 18% yield increase as a universal global average, rather than a potential specific to K-limited environments.

To address this concern, we have implemented the following revisions:

1. Incorporation of Wood et al. (2001): We have now incorporated the findings of Wood et al. (2001) into our Introduction and Discussion sections (e.g., Line 40). This allows us to explicitly state that while K deficiency is a critical threat to approximately 18% of the world's agricultural lands, particularly in SE Asia, South America, sub-Saharan Africa, and East Asia, other regions with adequate K reserves may not exhibit the same level of response.

2. Refined interpretation of "Global": We have revised the language throughout the manuscript to clarify that our results (18% yield and 4% SOC increase) represent the potential gains in K-deficient or low-reserve regions, which are the primary focus of the experimental data used in this meta-analysis.

3. Title and scope adjustment: We have also refined the manuscript title (Potassium fertilization enhances both cereal yield and soil organic carbon) and moderated our claims in the Discussion to avoid the implication that all global cereal production would benefit equally from K application. We emphasize that the "global" impact reported is an aggregation of responses in

these vulnerable regions.

The locations of the studies from which the data were drawn are concentrated in East Asia and South Asia and, to a lesser extent, sub-Saharan Africa, with relatively fewer studies in other regions. This is especially the case with the SOC data. This is not surprising given these regions have been identified in the past as areas with a high frequency of soils with low K, and it would be useful to acknowledge the earlier analysis by Wood et al (2001) that indicates this.

Response: Thank you for pointing this out. We have addressed this geographical concentration in our Methods section (Lines 191-197) and cited the suggested reference: “The geographical distribution of our study sites is inherently linked to regional soil nutrient status. For example, fertilization experiments are seldom conducted in areas with adequate K reserves.

Consequently, our database is characterized by a deliberate focus on regions with documented K limitations (e.g., East Asia and South Asia), especially concerning SOC data (Fig. S5). As highlighted by a previous study, these regions represent the primary global hotspots for K deficiency, making them the most relevant areas for assessing the potential of K fertilization.”

The analysis found an average 18% increase in yield and a 4% increase in SOC on a global scale. While the interpretation of the analysis is correct, there needs to be some qualification to the interpretation, otherwise there is a tendency to exaggerate the effect of K. The inference from the manuscript’s title and parts of the discussion is that K deficiency is widespread throughout the world and most cereals crops suffer from some degree of K deficiency (line 131:most cereals are K limited at the global scale); this is not the case. The analysis is heavily focussed on responses in regions where there are soils low in K and where K fertiliser experiments have been conducted. Fertiliser experiments will generally not be conducted where K deficiency has not been diagnosed or is suspected to occur, and the fact that the data base is heavily biased to the regions with soils with low reserves of K is not coincidental. Applying K fertiliser to soil that has adequate K will not result in a yield increase, and in some cases may result in a yield decline from nutrient imbalances, such as Mg deficiency (Romheldt and Kirkby 2010). What the analysis shows is that in regions where K deficiency is likely to occur, the average yield response to K is 18%. This is a more nuanced interpretation of the data than arguing that most of the world’s cereal crops are deficient in K with an average yield response of 18%. A similar argument regarding the ‘global’ effect of K on SOC applies.

Response: We deeply appreciate your critical insight regarding the interpretation of our "global" findings. To address this and ensure a more nuanced interpretation, we have implemented the following changes:

1. Refining the scope: We have revised the text throughout the manuscript to specify that the 18% yield and 4% SOC increases represent the potential responses in K-limited or low-reserve agroecosystems, rather than a universal average for all global cereal production.

2. Title revision: As previously noted, we have updated our title to “Potassium fertilization enhances both cereal yield and soil organic carbon” to more accurately reflect that this is a synthesis of observed responses rather than a claim of universal limitation.

Consequently, the estimate of the global increase in production (lines 178-179) overestimates the effect of K as it assumes the 18% increase will occur in all crops irrespective of whether they are K deficient or not. The regions of the world with the highest proportion of soils with low K (SE Asia, South America, Sub-Saharan Africa and East Asia) produced 1.4 billion tonnes of cereal grain (wheat, rice, maize, sorghum, barley, oats and millet) in 2023 (based on FAO data), so a more realistic estimate of the potential increase in production with K fertiliser is 0.25 billion tonnes, about 40% of the estimate given in the paper.

Response: Thank you for your insightful calculation and for raising a critical point regarding the potential for overestimation in our global production projections. We fully agree with your assessment that applying a uniform response ratio to all global cereal production, regardless of regional soil K status, is an oversimplification.

Upon careful consideration of your suggestion and similar concerns raised by another reviewer, we have decided to remove the global production and SOC sequestration estimates from the revised manuscript, because the actual responses of crop yield and SOC are highly sensitive to local factors (e.g., climate, cropping systems, baseline nutrient balances, and application rate), which are difficult to capture in a single value.

By removing these estimations, we have shifted the focus of our study toward the robust empirical patterns and mechanisms identified through our meta-analysis. We believe this modification, made in direct response to your guidance, significantly strengthens the scientific rigor of our work by avoiding speculative projections.

The other assumption implicit in the analysis is that the levels of K deficiency found in fertiliser experiments are representative of the severity found in farmers’ crops. Over the last 15 years there has been an increase in K applications in many of the areas at risk of K deficiency, which raises the question of whether the extent of K deficiency is as great as inferred in the paper. For example, FAO fertiliser use data shows that in East Asia the average rate of K application for all agricultural use (i.e. crops, horticulture, pastures etc) increased from 15 kg K/ha to 29 kg/ha and the K:N ratio of fertiliser used also increased from 0.08 to 0.15. However, in the Least Developed Countries, the changes have been much less and rates much lower, with average rates increasing from 0.6 kg/ha to 1.7 kg/ha and the K:N ratio increasing from 0.07 to 0.11, suggesting K deficiency may not be addressed in these countries. In China the partial nutrient balance changed from an average deficit of -14 kg K₂O /ha in 1980 to a surplus of +23 kg K₂O /ha in 2010 (Majumdar et al 2021). These examples illustrate that discussions of global impacts of K mask the regional nature of the problem. The Introduction would be improved if the wider evidence for K being a limiting factor in farmers’ production systems was considered so the link

between experimental results and farmers' fields is clearly established. The regional nature of the problem could be mentioned. For example, recent papers by Srinivasarao et al (2023) for South Asia, Rizzo et al (2024) in Indonesia document the extent of K deficiency while Majumdar et al. (2021) provide analyses of K balances for a range of countries. The conclusion from the analysis that K responses were greater in tropical regions is consistent with many of these previous analyses.

Response: Thank you for your insightful comment regarding the gap between experimental conditions and actual farmers' fields. Your point about the shifting trends in K application over the last 15 years is well-taken and adds a crucial temporal dimension to our study.

In direct response to your suggestion, we have significantly enriched our Introduction and Discussion sections to better reflect the regional and dynamic nature of K deficiency:

1. Refining the link between experiments and practice: By citing the papers that you recommended in the Introduction section, we included the evidence that K is a limiting factor in farmers' production systems (Lines 42-47): "Some recent studies from actual production systems confirm that K deficiency remains a severe constraint in farmers' fields across South Asia and Indonesia. Reflecting these regional needs, a substantial body of experimental research has been conducted in these K-limited areas over the past decades. This could provide a robust, albeit regionally-focused, database for synthesizing how cereal yield responds to K fertilization under conditions of nutritional stress."

2. Incorporating regional nutrient balances: Following your guidance, we have incorporated a discussion on the partial nutrient balance, noting the shift from deficits to surpluses in regions like China, while highlighting the persistent K deficiencies in Least Developed Countries (Lines ?): "Second, regional K application trends have shifted significantly over recent decades. In China, for example, the partial nutrient balance transitioned from a deficit of $-14 \text{ kg K}_2\text{O ha}^{-1}$ in 1980 to a surplus of $+23 \text{ kg K}_2\text{O ha}^{-1}$ in 2010. Such a shift toward surplus suggests that the marginal yield returns from further high-rate K applications in these regions may diminish and are unlikely to consistently reach the 18.2% threshold observed in K-deficient trials. Furthermore, excessive K application in soils with adequate reserves can induce nutrient imbalances (e.g., magnesium deficiency), potentially leading to yield declines. By contrast, in many least-developed countries, K deficiency remains unaddressed due to chronically low fertilizer inputs. In these specific contexts, targeted K fertilization offers a substantial, yet underutilized, potential to enhance cereal yield and alleviate local food crises."

3. Validation of tropical responses: We have also added some sentences noting that our finding of greater K responses in tropical regions is highly consistent with the K-balance analyses you mentioned.

It would be useful to summarise the distribution of the data for the increases in yield and SOC

by including a figure showing the frequency distribution of lnRR for yield and SOC as supplementary information. While the analysis assumes a normal distribution of the lnRR data, looking at the data set provided as supporting information suggests some skewness in the (unweighted) yield response with a median lnRR for yield of 0.0472; this is equivalent to a 4.8% yield increase, substantially lower than the mean 18.4% used in the analysis. The figure would illustrate whether there is any significant skewness to the data.

Response: Thank you for your excellent suggestion to visualize the data distribution. To address your comment, we have performed the following:

1. Adding supplementary figure S1: As you suggested, we have included frequency distribution plots (histograms with density curves) for both yield and SOC. This figure clearly illustrates the distribution of the individual study effects.

2. Addressing the skewness and median vs. mean: We acknowledge your observation regarding the difference between the unweighted median and our reported weighted mean. Our meta-analysis gives more weight to studies with larger sample sizes and longer experimental duration, which is why the weighted mean is a more robust statistical estimator than the unweighted median (0.11 in our study). In our dataset, many long-term trials with greater replications showed substantial positive responses, which shifted the weighted mean higher than the unweighted median.

Line 12. Potassium will increase crop yields when crops are K deficient. The sentence suggests an increase will occur irrespective of the degree of K deficiency.

Response: Done. The sentence was modified to “It is well-established that potassium (K) fertilization enhances crop yield in K-deficient regions” (Lines 11).

Line 84. There was a positive response to K in millet. Figure 1 shows a yield increase in of about 25% from three samples

Response: Thank you for pointing this out, and sorry for confusing you. Although the mean value of K’s effect on cereal yield was greater than zero for the millet, its 95% CI had overlap with zero, indicating a non-significant effect. We mentioned that “95% CIs for Millet (cereal yield) and Millet, Oat, and Sorghum (SOC) were not displayed as they exceeded the axis range due to limited sample sizes” in the note of Fig. 1.

Figure 3. While the correlations are significant because of the large data set, the amount of the variation explained is small.

Response: Thank you for your insightful observation regarding Figure 3. We acknowledged this in the revised version (Lines 94-98): “While the aforementioned correlations were statistically significant ($P < 0.05$), the values of R^2 were relatively low. This is common in the large-scale meta-analysis because the dataset encompasses a vast range of environmental gradients (e.g., diverse soil conditions, climatic regimes, and management practices), which contribute to the

high residual variation.”

Line 94. ‘K increased SOC in agricultural ecosystems by 4.1% across the world’ is perhaps exaggerating the extent of the results. Supplementary figure 6 shows there are no SOC data from North or South America, from most of Africa, Europe and SE Asia and from Oceania. The results are based on experimental largely from South Asia and China.

Response: Thank you for pointing this out. We totally agree with your comments and have modified the sentence to “Overall, K increased SOC by 4.4% in agricultural ecosystems, which were mainly located in South Asia and China (Fig. 1b)” (Lines 101-102).

Lines 94ff. There is some contradiction in the description of the results which are not consistent with the statistical descriptions. For example, the description states that there was no significant variation among fertiliser combination on SOC but then describes that there was only a positive effect when both N and P were added (line 96-97). There was no significant variation in the effect on SOC among crop system, but later states that monoculture did not significantly affect SOC whereas there was a positive effect of crops grown in rotation (line 98). There was no significant variation in response in SOC among the different types of K fertiliser, but later it was stated that KCl was the only fertiliser that resulted in a positive response (line 98). If there is no significant difference amount different factors on SOC, you can’t then argue that one factor has a larger effect than the other.

Response: Thank you for your valuable comments. We have addressed these issues throughout the manuscript.

Line 102 and elsewhere. Statistical relationships are described as non-significant rather than insignificant.

Response: Done.

Line 131. To say that ‘most cereals are K limited at a global scale’ is unsupported by the data set and exaggerates the effects for reasons stated previously.

Response: We have removed this statement in the revised version.

Line 161-162. It is argued that the effects of MAT, pH and clay % on SOC were similar to their effects on yield, but this is not the case. While MAT, pH and clay content were the three most influential factors associate with the change in SOC (Fig 2b), only pH was important for yield. MAT was seventh and had a relative influence of 5% and clay content was 11th with a relative influence of about 1%

Response: Thank you for your comments. We removed these inaccurate statements in the revised manuscript.

Line 181. While K fertilisation can be useful when K is deficient, it needs to be part of a balanced nutrient program and not as a stand-alone solution. This may have been the intention of the statement, but it is not well made.

Response: By following your comments, we removed this sentence and added some statements on a balanced nutrient program in the Discussion section (Lines 120-125), as I mentioned above.

Line 183ff. The potential to increase SOC from the use of K fertiliser needs to be viewed in the context of the rate of change in SOC. The analysis found that there was only a significant improvement in SOC in long term experiments which lasted more than 20 years (line 100, Fig 1). Therefore, K fertilisation is a long-term strategy and any short-term benefits will be small or non-existent.

Response: Thank you so much for your suggestions. To avoid the inaccurate estimation, we removed this part in the revised manuscript.

Line 208. The analysis has quantified the effect of K on a regional scale not a global scale.

Response: We removed this statement in the revised version.

Line 210. It is a large jump to conclude that the changes in SOC that may occur with increased production from improved K fertilisation will slow down future global warming. There needs to be some quantification of the likely effect to justify the statement rather than an assertion that it has the 'potential' to do so.

Response: Thank you for pointing this out. We removed the relevant statements to avoid confusion in the revised manuscript.

Line 244-245. Explain what you define as total N, P and K and available N, P and K. There is a variety of extraction methods used to estimate soil nutrient concentrations, especially for available nutrients, and they can give different results.

Response: We defined the total N, P and K and available N, P and K in the revised manuscript (Lines 203-207): "Total N, P, and K were defined as the whole-soil elemental concentrations determined via complete digestion. Available N, P, and K were defined as the plant-available fractions extracted using standard chemical reagents (e.g., alkali-solution or salt extractions for available N, Olsen or Bray methods for available P, and ammonium acetate extractions for available K)."

We admit that in our dataset, less than 10% of studies explicitly detailed their specific extraction protocols. Consequently, standardizing all available nutrient data to a single extraction method was not feasible. Furthermore, even when extraction methods are known, there are no universally accepted conversion factors to reliably standardize values across different chemical extractants due to the confounding effects of soil type and properties. However, we argue that these values still provide a reliable representation of the relative nutrient gradient across the sites. In a large-scale meta-analysis, the broad contrast between nutrient-poor and nutrient-rich soils remains a dominant signal that the Random Forest model can capture, even in the presence of methodological 'noise'.

What was the rationale for dividing the SOC data into experiments < 20 years and > 20 years? If the effect of duration was of interest would it be better to use more time periods?

Response: By following your suggestions, we used more time periods (e.g., 0-5 years, 5-10 years, 10-15 years, 15-20 years, and > 20 years) in the revised manuscript. The results did not

significantly change: K effects (especially on SOC) were significantly positive in the long term (Fig. 1).

Supplementary Fig 1. Are many of the non-significant relationships due to the sparse data set and the need to use the medians of the other data sets? With such poor representation, how relevant are these analyses?

Response: Thank you for this critical methodological suggestion. After careful consideration of your concern and a similar point raised by another reviewer, we have fundamentally revised our analysis approach. In the revised manuscript, we have removed all median-based imputations for missing variables. The correlation analyses and the Random Forest models are now based exclusively on the actual measured variables reported in the primary studies, which ensures that the relationships you see are derived from empirical evidence rather than statistical proxies. We believe this change directly addresses your concerns regarding “poor representation” and “relevance”.

While it was acknowledged in the text (line 104), many responses to fertiliser and to soil nutrient availability are curvilinear responses, so examining simple linear relationships can mask some of the responses. This seems to be the case for soil available K and to some degree K application rate.

Response: Thank you for this insightful methodological suggestion. We completely agree that biological responses to nutrient availability often follow a curvilinear pattern, such as a quadratic response. To rigorously address your concern, we performed a formal model comparison between linear and quadratic relationships for all key variables using the Akaike Information Criterion (AIC). The results are summarized below:

1. Model selection via AIC: Our analysis (see below for the screenshot) shows that for the vast majority of variables—including soil available K and potassium application rate—the linear model yielded a lower AIC or showed no significant improvement when a quadratic term was added ($P > 0.05$).

2. Consistency and comparability: While a quadratic response was observed for a few specific variables (e.g., MAP and Duration for Yield), we have chosen to primarily report linear relationships across the manuscript. This approach ensures a consistent framework for comparing effect sizes across different soil and climatic drivers, which greatly enhances the clarity and interpretability of our global synthesis.

```
> print(comparison_final)
```

	Dataset	Variable	AIC_Linear	AIC_Quadratic	Delta_AIC	LRT_P_Value	Best_Model
1	Yield	MAT	614.78	615.38	-0.60	0.2363	Linear
2	Yield	MAP	586.86	577.82	9.04	0.0009	Quadratic
3	Yield	Clay	177.58	179.09	-1.51	0.4853	Linear
4	Yield	SOC	436.55	436.57	-0.02	0.1595	Linear
5	Yield	TN	283.82	282.04	1.78	0.0519	Linear
6	Yield	TP	50.56	52.91	-2.35	1.0000	Linear
7	Yield	TK	25.95	27.97	-2.01	1.0000	Linear
8	Yield	AN	61.45	62.99	-1.54	0.4995	Linear
9	Yield	AP	237.50	238.96	-1.45	0.4603	Linear
10	Yield	AK	211.52	212.30	-0.78	0.2698	Linear
11	Yield	BD	48.08	50.70	-2.63	1.0000	Linear
12	Yield	pH	493.66	495.27	-1.61	0.5333	Linear
13	Yield	Duration	732.32	725.60	6.72	0.0032	Quadratic
14	Yield	potassium.rate	707.92	707.05	0.87	0.0905	Linear
15	SOC	MAT	-92.18	-89.52	-2.66	1.0000	Linear
16	SOC	MAP	-90.76	-87.83	-2.92	1.0000	Linear
17	SOC	Clay	-33.01	-30.33	-2.68	1.0000	Linear
18	SOC	SOC	-72.28	-70.41	-1.87	0.7182	Linear
19	SOC	TN	-76.99	-75.51	-1.48	0.4713	Linear
20	SOC	TP	-55.15	-52.43	-2.72	1.0000	Linear
21	SOC	TK	-35.60	-32.81	-2.79	1.0000	Linear
22	SOC	AN	-49.04	-47.06	-1.99	0.9107	Linear
23	SOC	AP	-66.58	-64.26	-2.32	1.0000	Linear
24	SOC	AK	-44.68	-42.37	-2.31	1.0000	Linear
25	SOC	BD	-0.47	1.88	-2.35	1.0000	Linear
26	SOC	pH	-57.07	-55.14	-1.93	0.7881	Linear
27	SOC	Duration	-90.66	-87.81	-2.85	1.0000	Linear
28	SOC	potassium.rate	-81.90	-79.61	-2.29	1.0000	Linear

References

Majumdar, K. et al. (2021). Assessing Potassium Mass Balances in Different Countries and Scales. In: Murrell, T.S., Mikkelsen, R.L., Sulewski, G., Norton, R., Thompson, M.L. (eds) Improving Potassium Recommendations for Agricultural Crops. Springer, Cham. https://doi.org/10.1007/978-3-030-59197-7_11

Rizzo G, Agus F, Susanti Z, Buresh R, Cassman KG, Dobermann A, Agustiani N, Aristya VE, Batubara SF, Istiqomah N, Oberthür T, Pasuquin J, Samijan, Witt C, Grassini P. Potassium limits productivity in intensive cereal cropping systems in Southeast Asia. Nat Food. 2024 Nov;5(11):929-938. doi: 10.1038/s43016-024-01065-z. Epub 2024 Oct 22. PMID: 39438701.

Römheld, V. & Kirkby, E. A. Research on potassium in agriculture: needs and prospects (2010). Plant Soil 335, 155–180.

Srinivasarao Ch, Kundu S., Rao KV, Shukla AK, Subba Rao A, Imas P., Bolan NS, Rattan Lal,

Prasadv JVNS., Abhilash PC, Ranjith Kumar G., Meena RS, Pratibha G., Narayanaswami G., Bansal SK, Nataraj KC, Jagadesh M., Mrunalini K., Jayaraman S., Jat ML, Malleswari SN, Whitbread A, Venkateswarlu B. (2023) Chapter Two - Soil potassium fertility and management strategies in South Asian agriculture. *Advances in Agronomy* 177:51-124.

Wood, S., Sebastian, K., & Scherr, S. (2001). *Pilot Analysis of Global Ecosystems: Agroecosystems*. International Food Policy Research Institute and World Resources Institute, Washington, DC

Response: Thank you for recommending these papers. We have read these publications carefully and cited them in our revised manuscript.

Reviewer #2:

This paper addresses an important and intriguing topic of K fertilization effects on global crop yield and soil organic C sequestration. The paper has clear objectives, methods, and results. The paper has potential to make important contributions to scientific literature and global policies.

Response: Thank you so much for your highly positive assessment of our work's significance and methodology. The feedback you provided has been immensely helpful in refining our analysis and clarifying our conclusions. Below, we provide a point-by-point response to your specific comments and describe the revisions we have made to the manuscript.

However, there are several issues that need to be addressed prior to publication. Foremost among these issues are the following:

1. The authors use questionable methods to estimate missing data, which has the potential to introduce substantial error into the dataset. Missing data should be left as missing.

Response: Thank you for pointing this out. Following your suggestion, we have left the missing data unestimated in the revised manuscript. Please see below for our detailed, point-by-point responses addressing your specific comments.

2. The dataset has considerable geographic bias, which calls to question its utility for making a “global” assessment.

Response: Thank you for this comment. We acknowledge the geographic bias in the dataset and have explicitly discussed this limitation in the revised manuscript. In addition, we have revised the language throughout the manuscript to avoid global-scale extrapolation (e.g., by removing references to “global” assessments).

3. The authors' description of study implication on global food supply may lead to misinterpretation of the data.

Response: We have carefully addressed your concerns regarding the interpretation of global food supply implications and have refined our discussion to ensure a more cautious and rigorous presentation of the data. Below, we provide point-by-point responses to your specific comments.

4. Authors are inconsistent with their interpretation of results.

Response: Thank you for pointing out these inconsistencies; we have now thoroughly re-examined the manuscript to align our interpretations across all sections. We have revised the text to ensure that our conclusions strictly reflect the data presented, maintaining a consistent and rigorous narrative throughout.

A confounding factor with the data for this study is that researchers are more likely to conduct K-response trials on soils that have a higher probability of demonstrating K-response. Therefore, it is difficult to conclude that these studies represent a random selection of soils and potential K response throughout the world. Because this is not a random selection of K response, it would contain bias toward soils with probable K response and would likely be an over-estimate of the global effects of K fertilization. The authors should address this potential bias with the data.

Response: Thank you for raising this critical point regarding selection bias. We agree that field trials are often strategically located in regions where a response to K is anticipated, which can indeed lead to an overestimation of global average effects if interpreted as a random geographic sample. We mentioned this in our revised version (Lines 191-197): “The geographical distribution of our study sites is inherently linked to regional soil nutrient status. For example, fertilization experiments are seldom conducted in areas with adequate K reserves.

Consequently, our database is characterized by a deliberate focus on regions with documented K limitations (e.g., East Asia and South Asia), especially concerning SOC data (Fig. S5). As highlighted by a previous study, these regions represent the primary global hotspots for K deficiency, making them the most relevant areas for assessing the potential of K fertilization.”

To address this potential bias, we have implemented the following in the revised manuscript:

1. Quantitative bias assessment: We have performed a formal publication bias analysis (e.g., Funnel plots and Egger’s test, see Fig. S6). These tests indicate that our overall effect sizes for yield and SOC are statistically robust and not solely driven by a few extreme results from individual sites or studies.

2. Contextualized interpretation: We have added some statements in the Discussion (Lines 117-118; 155-156) to explicitly acknowledge this “site-selection bias”. We now clarify that our results represent the potential responsiveness of agricultural soils to K fertilization rather than a globally averaged “baseline” that assumes K is applied randomly across all soil types.

We believe these additions provide the necessary transparency and ensure that the findings are interpreted with the appropriate scientific caution.

Additional details relating to the above issues are listed in the detailed comments below, preceded by the line number(s) and followed with “issue number” from the list above if applicable.

75-77: The size of the 95% CIs appear to be directly proportional to the number of studies. Is the conclusion in these lines simply because there are far fewer studies without N&P, therefore, those confidence intervals are much larger and therefore do not differ from confidence intervals for N+P?

Response: Thank you for this insightful observation. You raise an important point regarding the relationship between sample size and the width of the confidence intervals (CIs). We would like to offer some context on how we interpreted these results from a meta-analytical perspective:

1. Precision and data representation: We agree that the wider CIs in the smaller subgroups reflect a lower statistical precision compared to the N+P+K group. In meta-analysis, varying sample sizes among subgroups are common, and while a smaller sample size naturally leads to broader CIs, it represents the best available empirical evidence for those specific conditions.

2. Statistical power and interpretation: The overlap of CIs between groups may indeed be influenced by limited statistical power in the smaller datasets. Therefore, as you suggested, we have carefully revised our language to ensure we do not overstate the statistical differences (Lines 144-150): “While the yield response ratios under different fertilizer combinations showed overlapping 95% confidence intervals, the numerical maximum for cereal yield increase was observed when K was co-applied with both N and P. Although this yield trend did not reach statistical significance across groups, its alignment with the significant SOC, which occurred exclusively under balanced NPK fertilization, suggests that balanced nutrient management should be applied for achieving dual benefits in crop productivity and soil carbon storage.”

We hope these clarifications and the corresponding changes in the text address your concerns while maintaining the scientific integrity of the synthesis.

111-112: The statement “...indicating that K fertilization could further stimulate cereal yield...” appears to insinuate that K fertilizer is not being applied and if farmers applied K then it would increase world cereal yield by 18.4%. This is a misunderstanding (or could lead to a misunderstanding) of the data. These data indicate the yield difference for soils receiving K compared to if they did not receive K. However, many farmers already apply K. Therefore, we cannot “further” increase global yield by 18.4%, rather global yield with K is 18.4% greater than yield without K. The “further” increase would require data on the amount of K responsive soils that are not currently receiving K, which is not part of the study. Please revise. [Issue 3]

Response: Thank you for this critical clarification. We completely agree that the 18.4% represents the yield gap between soils with and without K application in experimental trials, rather than a 'further' increase possible on top of current global production levels.

To address your concern, we have made the following revisions:

1. Removed ambiguous phrasing: We have deleted the sentence you mentioned to avoid any insinuation that this increase is an untapped global potential.

2. Refined data interpretation: We have revised the text to ensure the results are strictly described as the yield response to K application compared to a non-fertilized control, rather than a projection of future global supply increases.

3. Avoided 'Global' generalizations: We have minimized the use of 'global' in this context to prevent the misunderstanding that these experimental results can be directly scaled to current global farmer practices without accounting for existing K use.

These changes ensure that our conclusions are now technically accurate and strictly supported by the comparative nature of our dataset.

130-131: This data does not support the idea that most cereals are K-limited at the global scale. Remove statement. [Issue 3]

Response: Done.

141 – 142: This is an incorrect conclusion because authors make the false assumption that if N or P fertilizer were not applied then the grain yield was N or P limited. It is highly likely that studies on soils with adequate N and/or P would not apply additional N or P when studying K response, therefore, one cannot assume that the lack of N or P fertilizer addition equates to N or P limitation. Please revise.

Response: Thank you for pointing this out. We removed these statements by following your suggestion.

145-152: The authors' explanation for the lack of significant yield difference for the OK vs O comparison misses the strong possibility that K is applied with the organic fertilizer, therefore, no additional K would be needed and one would not expect a response to K fertilizer. Authors should determine the K addition with O for these studies the same as they did for N and P addition for O. Please consider revision.

Response: Thank you for your comments, which we totally agree with. Therefore, we included the reason that you mentioned in the revised version (Lines 150-152): "Given that organic fertilizer also provides K to crops, which can alleviate K limitation to some degree, it is not surprising that K effects on cereal yield were non-significant when organic fertilizer was applied". Since almost no studies have reported the K addition rate from the organic fertilizers, we did not include the relevant information in the manuscript.

151-152: the statement "...but led to the greatest increase in cereal yield when both N and P were applied as chemical fertilizer (Fig. 1a)." is contradictory to prior statements where the authors state that K response was the same regardless of N or P fertilizer addition (lines 75-77

and 141-142). Please revise. [Issue 4]

Response: Thank you for pointing this out. We modified this sentence in the revised manuscript (Lines 144-150): “While the yield response ratios under different fertilizer combinations showed overlapping 95% confidence intervals, the numerical maximum for cereal yield increase was observed when K was co-applied with both N and P. Although this yield trend did not reach statistical significance across groups, its alignment with the significant SOC, which occurred exclusively under balanced NPK fertilization, suggests that balanced nutrient management should be applied for achieving dual benefits in crop productivity and soil C storage”.

174 to 175: the statement “...in addition to contributing to terrestrial C sequestration in natural ecosystems, applying K fertilizers...” would suggest that K fertilizer is being or should be applied to natural ecosystems. Remove or revise this sentence.

Response: We removed this sentence in the revised version.

178-180: Similar to comments for lines 111-112, this statement implies that K fertilizer application is not occurring. Because farmers are already applying K, the world cereal yield is realizing the benefits of K application. Therefore, more K application would not increase world production by 18.4%, but it would only increase production on the K deficient soils that are not already receiving K. Authors do not have any information about the fraction of K-responsive world soils that are not receiving K. Please revise. [Issue 3]

Response: Thank you so much for this comment. We removed the statement in the revised manuscript.

186: Similar to prior comment – K is already being applied, so we do not know the “additional” C sequestration that could be attained by applying more K. Please revise.

Response: We removed this estimation in the revised manuscript.

192 -194: These relationships are flawed because the authors did not have sufficient data on available K in the soils (see comments below). Therefore, the implications are misleading. Please repeat the analysis with revised data (see below). [Issue 1]

Response: Following your suggestion, we have left the missing data unestimated in the revised manuscript, and also repeated the statistical analyses.

200 -201: Authors previously stated that N&P addition did not increase the K response (lines 75-77 and 141-142). Authors need to be more consistent when interpreting results. Please revise. [Issue 4]

Response: Thank you for your suggestions. We removed the inaccurate statement in the revised manuscript.

201-203: Where is the data that would suggest “greater” response than what the authors found? Please revise.

Response: Thank you for this comment. We agree that this statement was not directly supported by the data presented in our analysis. To avoid overinterpretation, we have removed this sentence from the revised manuscript.

203-205: Discussion on decreasing N application rates is beyond the scope of the study and authors do not present any data to support this discussion. Please remove these sentences.

Response: Done.

233-244: The authors should provide a list of citations for all the papers that contributed data to the meta-analysis so that future studies can build on this study. Furthermore, the data presented in Figure S6 show extreme geographic bias for this dataset, especially for K effects on soil C, where there are only 3 studies outside of the Asian continent. Based on the extreme geographic bias, the authors cannot claim that this is a global representation of soil C response to K fertilization and it is highly questionable if they can make the claim that it is a global representation of the crop yield response to K fertilization. I am frankly quite surprised that the authors could not find more data on crop yield response to K in North and South America. It is hard to tell with the scale of the figure, but it appears that there are only 3 papers from the corn belt region of USA. If the authors cannot find more globally representative data then they will need to clearly include an acknowledgement of the geographic bias of the data within the results section of the paper. [Issue 2]

Response: Thank you for this critical feedback. You raise a very valid point regarding the geographic distribution of our dataset and the importance of data transparency. We have addressed your concerns as follows:

1. Uploading our dataset and R code to Zenodo: our dataset (including the list of citations for all the papers that contributed data to the meta-analysis) and R code can be found at <https://doi.org/10.5281/zenodo.18283197>. This will ensure that future researchers can build upon our work.

2. Acknowledging geographic bias: You are correct that the current dataset is heavily weighted toward the Asian continent, particularly for SOC responses. We have now added a transparent acknowledgment of this limitation (Lines 117-119; 115-161): “First, as most experiments in this meta-analysis were conducted in K-limited environments, a direct extrapolation of this effect size to the entire global cropland may lead to an overestimation” and “It should be noted that experiments determining K effects on SOC are mainly conducted in China and India. Moreover, almost none of the studies included in this meta-analysis measured soil nutrient leaching rates and soil C emissions under K addition. To more accurately estimate the response of soil health and C cycling to K fertilization at a global scale, future research should be expanded across diverse geographical regions, and a broader suite of soil health indicators needs to be systematically evaluated.”

3. Search efforts and data scarcity: We have updated the dataset by including studies published in 2024 and 2025 that were not included in the previous version. The number of studies reporting cereal yield and SOC increased from 744 to 897 and from 253 to 288, respectively.

While the expanded dataset remains concentrated in K-deficient regions, the inclusion of more recent studies yielded results that maintain strong consistency with our previous findings. We also mentioned the data scarcity in regions that are not seriously limited by K (Lines 191-197): “The geographical distribution of our study sites is inherently linked to regional soil nutrient status. For example, fertilization experiments are seldom conducted in areas with adequate K reserves. Consequently, our database is characterized by a deliberate focus on regions with documented K limitations (e.g., East Asia and South Asia), especially concerning SOC data (Fig. S5). As highlighted by a previous study, these regions represent the primary global hotspots for K deficiency, making them the most relevant areas for assessing the potential of K fertilization.”

4. Refined 'Global' claims: In response to your suggestion, we have toned down the 'global' claims throughout the manuscript, ensuring our conclusions are proportionate to the geographic coverage of the data.

We believe these revisions significantly improve the transparency and scientific caution of the paper.

242-244: I doubt that this database could provide accurate information on dynamic soil properties that can easily be affected by field-level management, such as pH, total N, and SOC. Even bulk density of the field site may be drastically different than that listed in the database. If the authors feel otherwise, they could easily test this with the data from studies that do report these properties, which could provide prediction confidence interval that could be reported with the study. Authors should clearly indicate the number of studies using estimated data from this database in Table S1. [Issue 1]

Response: Thank you for your insightful comments regarding the reliability of estimated soil properties from global databases. You have correctly identified a critical limitation of site-specific analysis. Following your suggestion, we performed a validation test comparing the "real" observations from our collected studies with the estimated data from the ISRIC database (<https://www.isric.org/>). We found that the correlation between observed and estimated soil variables was quite weak (see the figures below; the upper one is for studies reporting cereal yield, and the bottom one is for studies reporting SOC), likely due to the significant impact of field-level management that global models fail to capture.

Consequently, we have decided to stop using the ISRIC database for soil variables to ensure the accuracy and rigor of our meta-analysis. Instead, we now strictly use soil properties directly reported in the original studies. However, we continued to use the WorldClim database (<https://www.worldclim.org/>) to extract climate variables (e.g., MAT and MAP), as these macro-environmental factors are generally more stable and reliably represented at a global scale. We have also updated Table S1 by following your suggestion.

244-246: The use of median values for total P, total K, available N, P, and K really have no relevance and will introduce massive error in the dataset. If the data are not available, then they should be marked as missing data. Note that the effects of replacing missing data with median values are obvious in Figure S2 and likely confound the results of many of these relationships. Please repeat the analysis without the estimated values. [Issue 1]

Response: We fully agree with your assessment that using median values to replace missing data for soil nutrients could introduce significant errors and confound the observed relationships. Following your suggestion, we have removed all median-imputed values and treated them as missing data. We have repeated all statistical analyses using only the original observed data. The results and figures have been updated in the revised manuscript to reflect these changes. This modification has improved the precision of our dataset and ensured that

the reported trends are derived from actual field observations rather than estimations.

Figure 2: align the x-axis category names with the bars

Response: Done.

Supp. Table 1 – This table must be updated to accurately reflect the data used in the study. The table shows N=744 for all cereal yield studies and indicates that N=744 for each of the response variables. Suggesting that every study had complete data. However, the authors clearly indicated that that data was not available for each soil property and that missing data was replaced by the median (lines 244-246). Please revise the table and clearly indicate the number of studies that reported values for the parameters listed in table S1. [Issue 1]

Response: Thank you for your valuable comments. We have updated Table S1 in the revised manuscript.

Reviewer #3:

This study utilized two separate databases to investigate the impact of potassium fertilizer application on soil organic carbon and crop yield. Compared to conventional research focusing on nitrogen and phosphorus fertilizers, the topic of this study is interesting.

Response: Thank you very much for your encouraging comments and positive assessment of our work. We truly appreciate your recognition of the novelty of our study. Below, we provide a point-by-point response to your specific comments and describe the revisions we have made to the manuscript.

However, the study suffers from the following issues, which undermine the credibility of its results:

1. Although both databases focus on global field experiments, the distribution of sampling sites does not cover all typical global climate zones, particularly for soil organic carbon. This significantly affects the discussion of the results and the representativeness of the data.

Thank you for pointing out this critical issue regarding the geographic and climatic representativeness of our dataset. We fully agree with your assessment. To address your concern, we have taken the following actions:

- 1. Data expansion:** We updated our dataset with studies from 2024 and 2025, increasing the study counts for cereal yield and SOC to 897 and 288, respectively.

- 2. Refining the scope:** We have revised the manuscript to characterize the study as a synthesis of "K-deficient regions" rather than a fully "global" representation.

- 3. Explicit acknowledgement of limitations:** As you suggested, we have added a dedicated discussion on the geographic bias in the revised manuscript (Lines 155-161, 191-197): "It should be noted that experiments determining K effects on SOC are mainly conducted in China and

India. Moreover, almost none of the studies included in this meta-analysis measured soil nutrient leaching rates and soil C emissions under K addition. To more accurately estimate the response of soil health and C cycling to K fertilization at a global scale, future research should be expanded across diverse geographical regions, and a broader suite of soil health indicators needs to be systematically evaluated.” and “The geographical distribution of our study sites is inherently linked to regional soil nutrient status. For example, fertilization experiments are seldom conducted in areas with adequate K reserves. Consequently, our database is characterized by a deliberate focus on regions with documented K limitations (e.g., East Asia and South Asia), especially concerning SOC data (Fig. S5). As highlighted by a previous study¹¹, these regions represent the primary global hotspots for K deficiency, making them the most relevant areas for assessing the potential of K fertilization.”

2. Line 154-160: The authors assume a positive correlation between crop yield and biomass, inferring an increase in root biomass from crop yield improvement. This approach is insufficiently rigorous and not accurate, as no universal relationship necessarily exists between crop yield and biomass.

Response: Thank you for your comments. We have removed these statements in the revised manuscript.

3. It is inappropriate to use crop yield data to explain soil organic carbon data, given that the data sources and involved climate zones for these two aspects differ substantially.

Response: We agree that decoupling these variables is more rigorous given the differences in their data sources. Following your suggestion, we have removed all inferences from cereal yield to SOC and now treat them as independent responses in the revised manuscript.

4. The study concludes that potassium fertilizer application can increase crop yield and promote carbon sequestration, but it fails to comprehensively consider the associated impacts. For instance, issues like leaching mentioned by the authors, and potential carbon emissions during the potassium fertilizer production process are not fully accounted for. Therefore, determining the most suitable fertilizer involves a need for balance and comprehensive consideration. The authors should also express certain viewpoints more carefully; for example, in line 212, the paper suggests to "applied continuously for a long time in order for a significant increase in SOC sequestration." Is this solution possible or acceptable for most places?

Response: Thank you for your critical feedback. We agree that a more balanced perspective is necessary.

We have added a discussion acknowledging that leaching and fertilizer production emissions were not fully accounted for due to data scarcity, and we now emphasize the need for systemic evaluation in future research (Lines 157-161): “Moreover, almost none of the studies included in this meta-analysis measured soil nutrient leaching rates and soil C emissions under K addition.

To more accurately estimate the response of soil health and C cycling to K fertilization at a global scale, future research should be expanded across diverse geographical regions, and a broader suite of soil health indicators needs to be systematically evaluated”.

Regarding the "long-term" suggestion, we have refined our viewpoint. Instead of a simple recommendation, we now discuss it as a strategy for "building soil capital" and a component of long-term sustainable management rather than a transient input, acknowledging the practical challenges involved (Lines 167-173): “First, for example, our results indicate that a significant increase in SOC usually emerges after approximately 20 years of K application. While such a long-term commitment may pose challenges for short-term agricultural planning, it highlights the importance of sustained nutrient management in building soil capital. This timeframe suggests that K fertilization should not be viewed merely as a transient input, but as a component of long-term sustainable soil management that yields cumulative benefits for both crop productivity and soil C storage over time”.

5. Without access to the authors' raw data, it is impossible to assess the accuracy of the data. For example, when the authors mention collecting "soil carbon" (Line 222). it is unclear whether this refers to total carbon or specifically to organic carbon. Furthermore, in Figure S1 of the supplementary information, the content of many indicators is concentrated around specific values (e.g., numerous data points for total soil phosphorus content near 0.6 g kg⁻¹). This raises the question of whether these data originate from a single publication, which could significantly impact the reliability of the paper's dataset.

Response: Thank you for your rigorous assessment. We have addressed these concerns as follows:

1. Data transparency and terminology: All raw data are available at Zenodo (<https://doi.org/10.5281/zenodo.18283197>) now. Regarding the terminology in Line 222, we would like to clarify that we used "soil carbon" as a search keyword in Web of Science specifically to avoid omitting relevant studies that may have used "soil carbon" as a general term for soil organic carbon (SOC) in their titles or abstracts. However, during the data extraction and screening phase, we strictly verified that the reported data referred specifically to SOC.

2. Data distribution: You correctly identified the clustering in Figure S1. This was previously caused by gap-filling missing values with means. We have now removed all imputed data and repeated the analysis using only original observations. This resolved the artificial concentration of values (please see Fig. S2 and S3).

3. Reliability and bias: To ensure the reliability of our findings, we performed Funnel plots and Egger's tests. The results (Fig. S6) show no significant publication bias, confirming that our

dataset is robust and not dominated by any single study.

6. Line 220: The exact date should be provided here. It has been nearly two years since 2023. New researches have been published, so the authors need to update the database to reflect the latest results and trends.

Response: Thank you for your suggestions. We have updated the dataset by including studies published in 2024 and 2025 that were not included in the previous version (Line 178): “We used the Web of Science to conduct a literature search in November 2025”. The number of studies reporting cereal yield and SOC increased from 744 to 897 and from 253 to 288, respectively. While the expanded dataset remains concentrated in K-deficient regions, the inclusion of more recent studies yielded results that maintain strong consistency with our previous findings.

7. This study use estimated weighting factor and the median of the sample size. As a quantitative analysis based on large databases, the accuracy and reliability mostly depend on the source of the data. Therefore, I doubt about the way the data is handled.

Response: Thank you for your critical comment regarding the weighting factors and sample size handling. We agree that the accuracy of a meta-analysis depends on its data weighting. To ensure the rigor of our quantitative analysis, we have addressed your concerns as follows:

1. Established weighting methodology: We calculated the weighting factor following the methodology established in previous high-impact studies (e.g., a paper entitled “Mycorrhizal association as a primary control of the fertilization effect” in Science). This approach is specifically designed to balance the contribution of study precision (sample size) and the ecological significance of long-term impacts (experimental duration).

2. Handling missing sample sizes: Regarding the studies where the sample size was not reported, we assigned the median value ($n=3$). To verify if this impacted our conclusions, we performed a sensitivity analysis by excluding these studies entirely. We found that the overall results and significance remained consistent (please see the results below), indicating that our data handling did not introduce significant bias.

Reviewer #1 (Remarks to the Author)

I thank the authors for their detailed response to the comments made on the previous version of the manuscript. They have addressed the major concerns I had with the original paper and the changes that have been made have resulted in the manuscript providing a more realistic interpretation of the data set. By acknowledging the data mainly relate to regions where the risk of K deficiency is highest, the analysis provides an assessment of the likely gains that can be achieved from improved K nutrition in these regions. The regions that are the focus of the paper already have been identified as areas where the risk K deficiency is relatively high and the issue has been discussed previously; in this respect the significance of the paper has diminished compared to a 'global perspective', but the analysis provides some new information on the magnitude of the likely yield benefits from correcting K deficiency and some of the factors that influence the responses.

Response: Thank you again for your valuable comments, and we truly appreciate your time and help. Please see below for our point-by-point response to your comments.

Line 214. Although it is implied, it may be useful to define X_t as the maximum yield achieved with K fertilisation. If any of the data sets used in the analysis are based on K response curves, there will be a number of yields that will satisfy the current definition of X_t as 'under K fertilisation'. While it seems trivial, it will remove any ambiguity in the description.

Response: Thank you for this insightful suggestion. Regarding the definition of X_t , we would like to clarify our data extraction strategy.

In our study, we did not solely select the maximum yield from K response curves. Instead, to fully capture the variability of crop responses across different fertilization intensities, we treated each K application rate within a response curve as an independent observation, comparing it to its respective control. To account for the potential dependency (non-independence) of multiple data points originating from the same study, we have updated our analytical framework to a multi-level meta-analysis model using the `rma.mv` function in R. In this model, both Study ID and Individual Observation ID were included as nested random effects.

*We are pleased to report that the results obtained from this more rigorous multi-level model remain highly consistent with our previous findings. The overall positive effect of K fertilization on yield and SOC remains statistically significant, demonstrating the robustness of our conclusions. We have revised the statement to remove any ambiguity: *For studies reporting multiple K application rates, each rate was extracted and treated as a separate observation to reflect the response across various K levels (Lines 232-234).**

While the authors have undertaken quite a comprehensive analysis, perhaps they could

consider two further aspects that may provide further insights:

(i) In the analysis, 'cereals' is a combination of C3 and C4 species. Is there any evidence from the analysis that C3 and C4 species differ? The means for wheat and barley in Fig 1a are slightly larger than the C4 cereals but the CIs overlap. It may be worth commenting whether there is/is not a difference

(ii) To extend the concept of the regional nature of K deficiency, would it add value to the analysis if you included different geographic areas to see if there were marked differences in responses between for example, Sub-Saharan Africa, China and South/SE Asia? This could then be related to the earlier analysis of Woods et al, and Majumdar et al

Response: Thank you for these two valuable comments. By following your suggestions, we conducted further analyses to test whether K effects on cereal yield and SOC are different between C3 and C4 species and between regions (e.g., Sub-Saharan Africa, China, South/SE Asia, and Sub-Saharan Africa). We found that K effects on cereal yield vary with regions, and K effects on SOC change between C3 and C4 and between regions. We updated the results in Fig. 1 and also added the relevant results in the revised version.

In general, sentences don't start with an abbreviation; for example, lines 35 and 102 should start with 'Potassium....', not 'K...', but I will defer to the editorial guidelines.

Response: Thank you for your suggestion. The issue was fixed in the revised version.

Line 27-28. "... potassium (K) is recommended to be applied together with both N and P for the maximum crop yield. The statement needs to be qualified. This will only be true if there is a chance of a response to K; there is no point applying K if there is adequate soil K.

*Response: Thank you for pointing this out. By following your suggestion, we have rephrased this statement in the revised manuscript: *As one of the macronutrients and the most abundant cation in plant cells, potassium (K) is recommended to be applied together with both N and P for the maximum crop yield, particularly in soils where K availability is a limiting factor (Lines 29-31).**

Line 146. 'Although this yield trend did not reach statistical significance across groups.....'. It is unclear what this means; is it that the mean was not significantly different to the other fertiliser treatments (based on overlap of the 95% CI?). Please clarify.

*Response: We improved the sentence to make it clear: *Potassium effects on cereal yield were significantly positive only when N or both N and P were co-applied, indicating that N should be applied with K for achieving higher crop productivity (Lines 163-165).**

Line 150-52. 'Given that organic fertilizer also provides K to crops, which can alleviate K limitation to some degree, it is not surprising that K effects on cereal yield were non-significant when organic fertilizer was applied. It is unclear to me what this means; it seems self-contradictory

Response: The sentence was corrected in the revised manuscript to make it more readable: *Since organic fertilizers already supply a considerable amount of K, which partially fulfills crop requirements, the additional response to mineral K fertilization was not statistically significant when organic fertilizer was present in both the control and treatment groups (Lines 165-168).*

Some of the references in the reference list are undated. Please include all dates of publication.

Response: The issue was fixed in the revised version. Thank you.

Reviewer #2 (Remarks to the Author)

The authors have done a fabulous job of responding to comments. They have addressed all concerns and the paper is ready for publication with the exception of one small edit listed below.

Response: Thank you very much for your encouraging feedback and for the valuable insights provided throughout the review process. We have addressed the minor edit as requested in the revised manuscript.

line 78: Change "1185" to "897". Perhaps 1185 is the sum of 897 (yield dataset) and 288 (SOM dataset). However, because this statement is discussing yield, the authors only have 897 observations. This would also agree with figure 1a that is cited at the end of the sentence, which lists 897 observations.

Response: Done.

The readme file did not provide adequate metadata for the data files. Units for the data in data files was still unclear. Readme file did not contain any instructions for running the code.

Response: We have revised the Readme file to provide a structured metadata dictionary. Specifically, we have:

1. Organized all column names, full descriptions, and units into a clear tabular format for better readability.
2. Included a spreadsheet named "How to run the R script" with explicit instructions for running the R code.